
# Heterogeneous Kinetics of $H_2O$, $HNO_3$ and HCl on $HNO_3$ hydrates (α-NAT, β-NAT, NAD) in the range 175-200 K

**Riccardo Iannarelli[1,2] and Michel J. Rossi[1]**

[1]Laboratory of Atmospheric Chemistry (LAC), Paul Scherrer Institute (PSI), CH-5232 PSI

Villigen, Switzerland; [2]New address: Safety, Prevention and Health Domain, RI DSPS-SCC ,

Station 6, Ecole Polytechnique Fédérale de Lausanne (EPFL), CH-1015 Ecublens,

Switzerland.

Correspondence to: M. J. Rossi (michel.rossi@psi.ch)

**Abstract**

Experiments on the title compounds have been performed using a multidiagnostic stirred-flow
reactor (SFR) in which the gas- as well as the condensed phase has been simultaneously
investigated under stratospheric temperature conditions in the range 175-200 K. Wall
interactions of the title compounds have been taken into account using Langmuir adsorption
isotherms in order to close the mass balance between deposited and desorbed (recovered)
compounds. Thin solid films at 1 μm typical thickness have been used as a proxy for
atmospheric ice particles and have been deposited on a Si window of the cryostat where the
optical element was the only cold point in the deposition system. FTIR absorption
spectrometry in transmission as well as partial and total pressure measurement using residual
gas MS and sensitive pressure gauges have been employed in order to monitor growth and
evaporation processes as a function of temperature using both pulsed gas admission and
continuous monitoring under SFR conditions. Thin solid $H_2O$ ice films were used as the
starting point throughout, with the initial formation of α-NAT followed by the gradual
transformation of α- ➔ β-NAT starting at 185 K. NAD was formed at once at somewhat
larger partial pressures of $HNO_3$ deposited on pure $H_2O$ ice. In contrast to published reports
the formation of α-NAT proceeded without prior formation of an amorphous $HNO_3/H_2O$
layer and always resulted in β-NAT. For α- and β-NAT the temperature dependent
accommodation coefficient $\alpha(H_2O)$ and $\alpha(HNO_3)$, the evaporation flux $J_{ev}(H_2O)$ and
$J_{ev}(HNO_3)$ and the resulting saturation vapor pressure $P_{eq}(H_2O)$ and $P_{eq}(HNO_3)$ were
measured and compared to binary phase diagrams of $HNO_3/H_2O$ in order to afford



thermochemical control of the kinetic parameters. The resulting kinetic and thermodynamic
parameters of activation energies for evaporation ($E_{ev}$) and standard heats of evaporation
$\Delta H_{ev}^{0}$ of $H_2O$ and $HNO_3$ for α- and β-NAT, respectively, led to an estimate for the relative
standard enthalpy difference between α- and β-NAT of -6.0 ± 20 kJ/mol in favor of β-NAT,
as expected, despite a significantly larger value of $E_{ev}$ for $HNO_3$ in α-NAT. This in turn
implies a substantial activation energy for $HNO_3$ accommodation in α- compared to β-NAT
where $E_{acc}(HNO_3)$ is essentially zero. The kinetic ($\alpha(HCl)$, $J_{ev}(HCl)$) and thermodynamic
($P_{eq}(HCl)$) parameters of HCl-doped α- and β-NAT have been determined under the
assumption that HCl adsorption did not significantly affect $\alpha(H_2O)$ and $\alpha(HNO_3)$ as well as
the evaporation flux $J_{ev}(H_2O)$. $J_{ev}(HCl)$ and $P_{eq}(HCl)$ on both α- and β-NAT are larger than
the corresponding values for $HNO_3$ across the investigated temperature range but significantly
smaller than the values for pure $H_2O$ ice. This means that once contaminated with HCl the
"impurity" HCl will persist along with $HNO_3$ upon complete evaporation of the atmospheric
ice particle. We comment on recent laboratory results involving the $HNO_3/H_2O$ system using
Chilled Mirror Hygrometers (CMH) in light of the present kinetic results.
**1    Introduction**
Heterogeneous processes taking place on ice clouds in the Upper Troposphere (UT) or on
Polar Stratospheric Clouds (PSC's) in the Lower Stratosphere (LS) have, since a long time,
been recognized as one of the major ozone depleting mechanism (Solomon et al., 1986).
PSC's consist of either particles of crystalline nitric acid trihydrate (NAT) (type Ia), ternary
$H_2SO_4/HNO_3/H_2O$ supercooled solutions (type Ib) or pure $H_2O$ ice (type II) (Zondlo et al.
2000) and are formed during the polar winter season when temperatures are sufficiently low
in order to allow $H_2O$ supersaturation that ultimately leads to cloud formation in the dry
stratosphere subsequent to ice nucleation (Peter, 1997).
Ozone is depleted during the Arctic and Antarctic spring season after unreactive chlorine
reservoir compounds, $ClONO_2$ and HCl, are converted into molecular chlorine and rapidly
photolyze into active atomic chlorine during the spring season (Solomon, 1990). The presence
of PSC's enables heterogeneous chemical reactions such as Reaction (R1), which represents
one of the most efficient stratospheric heterogeneous reactions (Friedl et al, 1986; Molina et
al., 1985, 1987):



$$ClONO_2(g) + HCl(s) \rightarrow Cl_2(g) + HNO_3(s) \qquad (R1)$$
Reaction (R1) is orders of magnitude faster than the corresponding homogeneous gas phase
process (Molina et al., 1985) and the most important chlorine-activating reaction in the polar
stratosphere. The contribution to ozone destruction from Reaction (R1) is twofold: first, the
released molecular $Cl_2$ rapidly photolyzes into atomic Cl establishing a cycle of $O_3$
destruction and, second, the overall removal of nitrogen oxides from the gas phase by
entrapment of $HNO_3$ in the ice, facilitates $O_3$ destruction through a gas phase catalytic cycle
similar to the one reported in Reactions (R2)-(R4):
$$X + O_3 \rightarrow XO + O_2 \qquad (R2)$$
$$\underline{XO + O \rightarrow X + O_2} \qquad (R3)$$
$$\text{net: } O_3 + O \rightarrow O_2 + O_2 \qquad (R4)$$
where X is H, OH, NO, Cl or Br leading to $HO_x$, $NO_x$, $ClO_x$ and $BrO_x$ catalytic cycles,
respectively.
Reaction (R1) increases the concentration of $HNO_3$ in the condensed phase and when PSC
particles become sufficiently large and fall out of the stratosphere, active nitrogen is
permanently removed through denitrification which has been observed in the field (Fahey et
al., 2001). Lower concentrations of nitrate owing to the absence of $HNO_3$ inhibit reactions
such as Reaction (R5):
$$ClO + NO_2 + M \rightarrow ClONO_2 + M \qquad (R5)$$
which form reservoir species with longer atmospheric residence times.
The study of $HNO_3$ interaction with ice in the temperature and pressure ranges typical of the
UT/LS is crucial in order to understand the de-nitrification process initiated by reaction (R1)
and its effectiveness in the overall ozone destruction mechanism. To this purpose, many
research groups (Voigt et al., 2000, 2005; Fahey et al., 2001; Schreiner et al., 2003; Gao et al.,
2004; Höpfner et al., 2006) have studied the composition of PSC's using both *in situ* and
remote sensing techniques both in the Arctic as well as above Antarctica. A balloonborne
experiment at first detected non-crystalline $HNO_3$ hydrates (Schreiner et al., 1999), later both
ballonborne (Voigt et al., 2000; Schreiner et al., 2003) and aircraft campaigns (Voigt et al.,
2005) obtained unambiguous proof of the presence of crystalline $HNO_3$ hydrates (NAT) at
altitudes between 18 and 24 km in the Arctic. The presence of β-NAT, through the





identification of type Ia PSC's, has been unambiguously confirmed by Höpfner et al. (2006)
using the MIPAS instrument on a satellite platform by comparison of measured limb-emission
spectra of polar stratospheric clouds with measured optical constants in the region of the
symmetric $NO_3$ peak at $\nu_2 = 820$ cm$^{-1}$.
The existence of several crystalline hydrates of nitric acid has been confirmed for several
years. Hanson and Mauersberger (1988) have identified two stable hydrates, namely, nitric
acid monohydrate (NAM, $HNO_3 \bullet H_2O$) and nitric acid trihydrate (NAT, $HNO_3 \bullet 3H_2O$) by
measuring the vapour pressure of mixtures of ice and $HNO_3$. The observed vapour pressures
of $HNO_3$ and $H_2O$ in the polar atmosphere indicate that only NAT may be of atmospheric
importance. Several distinct crystalline hydrates of $HNO_3$ have been found by Ritzhaupt and
Devlin (1991) in their work examining the infrared absorption spectrum of thin film samples.
By depositing the equilibrium vapours of aqueous $HNO_3$ solutions of different concentrations
at 293 K they observed nitric acid dihydrate (NAD, $HNO_3 \bullet 2H_2O$), NAM and NAT. Ji and
Petit have performed an in-depth and ground-breaking investigation on the thermochemical
properties of NAD (Ji and Petit, 1993).
Tolbert and coworkers have also reported infrared absorption spectra of NAM, NAD and
NAT in a series of studies. Tolbert and Middlebrook (1990) have co-condensed calibrated
mixtures of $H_2O/HNO_3$ vapours onto a cold support and assigned the absorption spectra of the
growing thin films to nitric acid hydrates (NAM, NAD or NAT) according to the ratio of the
dosing gases. Koehler et al. (1992) have observed the Fourier transform infrared (FTIR)
absorption spectra in transmission of nitric acid hydrate thin films and measured their
composition using temperature-programmed desorption (TPD). They confirmed the
previously assigned spectra of NAD and NAM. They were also the first to observe two
distinct structures of NAT: a low-temperature and metastable structure they called α-NAT
and a thermodynamically stable high-temperature structure named β-NAT. Middlebrook et al.
(1992) observed that NAD consistently converts to β-NAT when exposed to $H_2O$ partial
pressures typical of the stratosphere and therefore proposed that NAD is also metastable
under stratospheric conditions.
Several other groups have investigated the structure of nitric acid hydrates and published
absorption spectra of both α-NAT and β-NAT in the mid-IR range, using grazing incidence
Reflection Absorption IR spectroscopy (RAIRS) (Zondlo et al., 1998; Zondlo et al., 2000;



Ortega et al., 2003; Ortega et al., 2006; Herrero et al., 2006; Escribano et al., 2007) and FTIR
in transmission (Tso and Leu, 1996; Martin-Llorente et al., 2006; Ortega et al., 2006).
The study of the phase diagram of the system $H_2O/HNO_3$ showed evidence that NAD may as
well occur in at least two different structures (Beyer and Hansen, 2002). The two structures
are both metastable and convert into NAM and NAT depending on experimental conditions.
Grothe et al. (2004) also reported polymorphism of NAD where the formation of α-NAD or
β-NAD strongly depended on the temperature of crystallization.
Compared to the molecular properties of the nitric acid hydrates knowledge of the kinetic
parameters of trace gases interacting with $HNO_3$ hydrates is scarce. Middlebrook et al. (1992)
have used time-dependent FTIR monitoring of the optical density of growing NAT films
during deposition to measure the uptake of $H_2O$ and $HNO_3$ on NAT. They reported a value of
$\gamma_{NAT}(HNO_3) > 0.4$ for $HNO_3$ net uptake (γ) on NAT at T = 197 K whereas the range $2.0\times10^{-3}$
$\leq \gamma_{NAT}(H_2O) \leq 1.0\times10^{-2}$ is reported for $H_2O$, respectively. The range measured for $\gamma_{NAT}(H_2O)$
corresponds to the $HNO_3$ pressure used during the deposition. Using evaporation experiments
in a slow-flow reactor Biermann et al. (1998) measured the accommodation coefficient of
$H_2O$ on β-NAT substrates, $\alpha_{\beta-NAT}(H_2O)$, from the thickness of the substrate measured using
FTIR absorption. They found no temperature dependence, reporting lower limiting values of
$\alpha_{\beta-NAT}(H_2O) = (2.2 - 6.0)\times10^{-2}$ in the temperature range 192-202 K.
Delval and Rossi (2005) have used a multidiagnostic flow reactor, similar to the one used in
this work, coupled with a quartz crystal microbalance (QCMB) for the measurement of the
evaporation rate of $H_2O$ from α-NAT and β-NAT thin films. They reported a positive
temperature dependence of $\alpha_{\alpha-NAT}(H_2O)$ and a negative temperature dependence of
$\alpha_{\beta-NAT}(H_2O)$ in the temperature range 179-208 K.
Hanson (1992) also measured the uptake coefficient of $HNO_3$ on NAT using a cold coated-
wall flow tube with $HNO_3$ deposited on ice condensed on the cold flow tube walls and
reported $\gamma_{NAT}(HNO_3) > 0.3$. A rapid uptake was observed which decreased as the surface
coverage or dose of $HNO_3$ increased. Furthermore, the observed steady state partial pressure
of $HNO_3$ over the ice substrate is about a factor of 5 higher than the $HNO_3$ vapor pressure
over NAT and thus indicates that no hydrate was actually formed during the experiments.
Therefore, the observed uptake has most likely to be attributed to uptake on other cold
surfaces in the flow reactor.



Reinhardt et al. (2003) reported $\gamma_{NAT}(HNO_3) = 0.165$ in the temperature range 160 to 170 K.
They used a slow flow reaction cell coupled with DRIFTS (Diffuse Reflectance Infrared
Fourier Transform Spectroscopy) for the detection of adsorbed species and downstream FTIR
for the detection of gas phase $HNO_3$.
Hynes et al. (2002) observed continuous uptake of $HNO_3$ on water-ice films below 215 K and
time dependent uptake above 215 K, with the maximum uptake $\gamma_{ice}(HNO_3)$ decreasing from
0.03 at 215 K down to 0.006 at 235 K. They also observed that the uptake of HCl at 218 K on
ice surfaces previously dosed with $HNO_3$ is reversible. Furthermore, the adsorption of $HNO_3$
on ice surfaces which contained previously adsorbed HCl indicates that HCl is displaced from
surface sites by $HNO_3$.
In this work, the results for the kinetics of $H_2O$ and $HNO_3$ gas interacting with solid $HNO_3$
hydrates will be presented. The independent measurement of the rate of evaporation $R_{ev}$
[molec $s^{-1}$ $cm^{-3}$] and the accommodation coefficient $\alpha$ of $H_2O$ and $HNO_3$ on $\alpha$- and $\beta$-NAT
substrates is performed using a combination of steady state and real time pulsed valve
experiments. Results on the kinetics of HCl on $HNO_3$ hydrates will also be presented. All
experiments reported in this work have been performed using a multidiagnostic stirred flow
reactor (SFR), which has been described in detail before (Chiesa and Rossi, 2013; Iannarelli
and Rossi, 2014). In addition, all experiments have been performed under strict mass balance
control with a knowledge on how many molecules of $HNO_3$, HCl and $H_2O$ were present in the
gas vs. the condensed phase (including the vessel walls) at any given time. These experiments
have been described by Iannarelli and Rossi (2015).

## 2   Experimental Apparatus and Methodology


### 2.1   Experimental Apparatus and Growth Protocols


Figure 1 shows a schematic of the reactor used in this work with the experimental diagnostic
tools and Table 1 reports its characteristic parameters. Briefly, it consists of a low-pressure
stainless steel reactor, which may be used under static (all valves closed) or stirred flow (gate
valve closed, leak valves open) conditions. We use absolute total pressure measurement and
calibrated residual gas mass spectrometry (MS) to monitor the gas phase and FTIR
spectroscopy in transmission for the condensed phase. Thin solid films of up to 2 µm
thickness are grown on a temperature controlled Si substrate and an average of 8 scans are



recorded at 4 cm$^{-1}$ resolution in the spectral range 700-4000 cm$^{-1}$ at typical total scan time of
45-60 s.
The 1" Si window is the only cold spot in the reactor exposed to admitted gases and therefore
the only place where gas condensation occurs. This allows the establishment of a 1:1
correspondence between the thin film composition and the changes in the gas partial pressures
in the reactor. Experimental proof of mass balance has previously been reported for this setup
(Delval et al., 2003; Chiesa and Rossi, 2013; Iannarelli and Rossi, 2014).
The introduction of $HNO_3$ in the system forced us to slightly modify the inlet system used
previously (Iannarelli and Rossi, 2014) in order to take into account the fact that $HNO_3$ is an
extremely "sticky" molecule that interacts with the internal surfaces of the reservoir vessel of
the inlet system as well as with the reactor walls of the SFR (Iannarelli and Rossi, 2015).
Similarly to the case of HCl and $H_2O$ (Iannarelli and Rossi, 2014) we have described the
$HNO_3$ interaction with the reactor walls using a Langmuir adsorption isotherm and
determined the concentration of $HNO_3$ in the ice sample after calibration of $HNO_3$ following
the methodology described in Iannarelli and Rossi (2015). Table 2 reports the values of the fit
parameters of the Langmuir adsorption isotherms for all the gases interacting with the
stainless steel (SS304) internal surfaces of the SFR. Binary combinations of $HNO_3/H_2O$ and
$HCl/H_2O$ have been used to describe the interaction of the acidic probe gas with the vessel
walls in the presence of $H_2O$ vapor.
The protocol for the growth of α-NAT, β-NAT and NAD thin films has also been described
in Iannarelli and Rossi (2015). Briefly, the protocol for either hydrate always starts with the
growth of pure ice: the chamber is backfilled under SFR conditions with water vapor at flow
rates between $5 \times 10^{15}$ and $10^{16}$ molec s$^{-1}$, corresponding to a partial pressure of $H_2O$, $p(H_2O)$
between 4.7 and $9.4 \times 10^{-4}$ Torr (both apertures open), with the Si substrate held at temperature
in the range 167 to 175 K. The pure ice film grows on both sides of the Si substrate to a
thickness of typically 1 μm and the $H_2O$ flow is halted (Iannarelli and Rossi, 2014). The
temperature of the support is then set to the value used for the growth of the desired $HNO_3$
hydrate at a typical rate of ±0.3 K min$^{-1}$.
The growth protocols for α-NAT and NAD are similar and start after the deposition of a pure
ice film: the temperature of the Si substrate is held in the range 180 to 185 K for α-NAT and
at 168 K for NAD. The sample is exposed for approximately 10 min at SFR conditions to
$HNO_3$ vapor at flow rates in the range 3 to $7 \times 10^{14}$ molecule s$^{-1}$ for α-NAT and $9 \times 10^{14}$



molecule s$^{-1}$ for NAD. The typical total dose of HNO$_3$ admitted into the reactor is 2 to 3×10$^{17}$
molecules and 4×10$^{17}$ molecules for α-NAT and NAD, respectively, with almost all of it
adsorbed onto the ice film. In both cases, we observe the formation of a new phase after
approximately 5 min of exposure as shown in the change of the FTIR absorption spectrum.
The present experimental conditions seem to show that no nucleation barrier is present for α-
NAT and NAD growth, in agreement with previous works (Hanson, 1992; Middlebrook et al.,
1992; Biermann et al., 1998). In contrast, Zondlo et al. (2000) have shown that crystalline
growth occurs via an intermediate stage of supercooled H$_2$O/HNO$_3$ liquid forming over ice.
After exposure the temperature of the substrate is set to the desired value for the kinetic
experiments on α-NAT or NAD as a substrate.
The protocol for the growth of β-NAT is different compared to NAD and α-NAT hydrates as
it only starts after the growth of an α-NAT film. After the HNO$_3$ flow has been halted, the α-
NAT/ice system is set to static conditions and the temperature increased to 195 K. During the
temperature increase the α-NAT film converts to β-NAT as shown by means of FTIR
spectroscopy (Koehler et al., 1992; Iannarelli and Rossi, 2015), and once the conversion is
completed the temperature is set to the desired value to start the kinetic experiments using β-
NAT as substrate. Typical growth protocols under mass balance control showing both the
FTIR transmission as well as the corresponding MS signals of HNO$_3$ as a function of
deposition time have been published previously (Iannarelli and Rossi, 2015).
In all samples used for this work, we never have a pure HNO$_3$ hydrate because we always
operate under conditions of excess of ice. Excess ice has been shown to have a stabilizing
effect on both α-NAT and β-NAT (Weiss et al., 2016) and in all our experiments the presence
of excess ice has been confirmed from FTIR spectra (Iannarelli and Rossi, 2015).
**2.2   Experimental Methodology**
The experimental methodology used in this work is an extension of the methodology reported
in Iannarelli and Rossi (2014) where the combination of real-time pulsed valve and steady
state experiments allowed the independent measurement of the rate of evaporation R$_{ev}$ [molec
s$^{-1}$ cm$^{-3}$] and the accommodation coefficient α of HCl and H$_2$O on crystalline and amorphous
HCl hydrates.
For each gas X (X = H$_2$O, HNO$_3$, HCl) admitted into the reactor in the presence of ice, the
following flow balance equation holds at steady state:



$$F_{in}(X) + F_{des}(X) + F_{ev}(X) = F_{SS}(X) + F_{ads,w}(X) + F_{ads,ice}(X) \qquad (1)$$
All terms are flow rates in molec s$^{-1}$: $F_{in}$ is the flow rate of molecules admitted into the
reactor, $F_{des}$ the flow rate of molecules desorbing from the reactor walls, $F_{ev}$ the flow rate of
molecules evaporating from the ice surface, $F_{SS}$ the flow rate of molecules effusing through
the leak valve into the MS chamber, $F_{ads,w}$ the flow rate of molecules adsorbing onto the
reactor walls and $F_{ads,ice}$ the flow rate of molecules adsorbing onto the ice film.
Under the assumption that the adsorption onto the walls may be described as a Langmuir-type
adsorption, Eq. (1) may be expressed as follows for a gas X:

$$V \cdot R_{in}(X) + N_{TOT} \cdot k_{des,w}(X) \cdot \theta + V \cdot R_{ev}(X) =$$

$$= V \cdot R_{SS}(X) + S_w \cdot \frac{\alpha_w(X) \cdot \bar{c}}{4}(1 - \theta)[X]_{SS} + S_{film} \cdot \frac{\alpha_{film}(X) \cdot \bar{c}}{4}[X]_{SS} \qquad (2)$$
where V is the reactor volume in cm$^3$, $R_{in}(X)$ the rate of molecules X admitted in the chamber
in molec·s$^{-1}$·cm$^{-3}$, $N_{TOT}$ the total number of molecules X adsorbed onto the reactor walls,
$k_{des,w}(X)$ the desorption rate constant from the reactor walls in s$^{-1}$, $\theta$ the fractional surface
coverage in terms of a molecular monolayer, $R_{ev}(X)$ the rate of evaporation of X from the ice
in molec·s$^{-1}$·cm$^{-3}$, $R_{SS}(X)$ the rate of effusion through the leak valve in molec·s$^{-1}$·cm$^{-3}$, $S_w$ and
$S_{film}$ the surfaces of the reactor walls and the thin film in cm$^2$, $\alpha_w(X)$ and $\alpha_{film}(X)$ the
accommodation coefficients of X on the walls and on the thin film, $[X]_{SS}$ the concentration at
steady state in molec cm$^{-3}$ and $\bar{c}$ the mean thermal velocity of a molecule in cm·s$^{-1}$,
respectively. The mathematical derivation of Eq. (2) may be found in Supplement B of
Iannarelli and Rossi (2014).
Pulsed valve (PV) experiments and Langmuir adsorption isotherms have been used in order to
measure $k_{des,w}(X)$ and $\alpha_w(X)$ (Iannarelli and Rossi, 2014), leaving only two unknown
parameters in Eq. (2): $R_{ev}(X)$ and $\alpha_{film}(X)$. The Langmuir adsorption isotherms are shown in
Figure S1 of Supplement A whereas the parameters for the best fit are reported in Table 2.
In the case of $H_2O$, once the selected substrate has been grown according to the protocol
briefly described above, the film is set to a chosen temperature. After steady state conditions
are established, a series of $H_2O$ pulses are admitted into the reactor. The exponential decay of
the MS signal at m/z 18 ($k_d$) is given by the sum of the measured $k_{esc}$, the adsorption rate
constant on the walls ($k_w$) and the adsorption rate constant ($k_c$) onto the ice, namely $k_d = k_{esc} +$





$k_w + k_c$, in the aftermath of a pulse. The accommodation coefficient $\alpha_{film}(H_2O)$ may be then
calculated according to Eq. (3):
$\alpha_{film}(H_2O) = \dfrac{k_c(H_2O)}{\omega(H_2O)}$ (3)
where $\omega(H_2O)$ is the calculated gas-surface collision frequency in s$^{-1}$ and is reported in Table

280   1.

The steady state MS signal established before the pulse series represents the calibrated flow
rate of molecules effusing through the leak valve, $F_{SS}(H_2O)$, in Eq. (1) and it may be used to
calculate the concentration at steady state $[X]_{SS}$ according to Eq. (4):
$[X]_{SS} = \dfrac{F_{SS}(X)}{k_{esc}(X)V}$ (4)
where $k_{esc}(X)$ is the effusion rate constant of gas X out of the reactor in s$^{-1}$ (see Table 1).
Finally, $[X]_{SS}$ is used to calculate $R_{ev}(X)$ using Eq. (2).
Subsequently, the film is set to a higher temperature, $F_{SS}(H_2O)$ is recorded and a series of $H_2O$
pulses applied to the same ice sample. This experimental protocol has been repeated for each
measured point in the temperature interval of interest.
Under the present experimental conditions, PV experiments of $HNO_3$ leading to transient
supersaturation of $HNO_3$ are hampered by excessive pulse broadening, most probably owing
to the strong adsorption of $HNO_3$ on ice and the stainless steel vessel walls that makes the
observation and interpretation of a $HNO_3$ pulse difficult for low doses in the presence of ice.
In this case the advantage of the PV technique as a real-time method of observation is lost.
Therefore, in order to measure the kinetics of $HNO_3$ gas in the presence of α-NAT, β-NAT
and NAD ice films we have used the two-orifice method first described by Pratte et al. (2006).
It has been modified to take into account the interaction of $HNO_3$ with the internal walls of
the SFR. The two-orifice method has also been used to measure the kinetics of $H_2O$ on $HNO_3$
hydrates in order to compare these results with the results of PV experiments for $H_2O$.
The two-orifice (TO) method allows the separation of the rate of evaporation $R_{ev}(X)$ and the
condensation rate constant $k_c(X)$ of a gas X by choosing two different escape orifices and
measuring the corresponding value of concentration $[X]_{SS}$ at steady state of gas X inside the
reactor. By alternatively opening the small orifice (S) and both orifices (M) (see Figure 1),
two steady state equations hold for a probe gas X which are reported in Eqs. (5) and (6) taking
into account the interaction with the reactor walls:





$\quad R_{ev}(X) + \frac{N_{TOT}}{V} \cdot k_{des,w}(X) \cdot \theta = (k_c(X) + k_{esc}^S(X)) \cdot [X]_{SS}^S + \frac{k_w(X)}{V} \cdot (1 - \theta) \cdot [X]_{SS}^S$
$\qquad$ (5)
$\quad R_{ev}(X) + \frac{N_{TOT}}{V} \cdot k_{des,w}(X) \cdot \theta = (k_c(X) + k_{esc}^M(X)) \cdot [X]_{SS}^M + \frac{k_w(X)}{V} \cdot (1 - \theta) \cdot [X]_{SS}^M$
$\qquad$ (6)
where the superscript indexes indicate small orifice only (S) or both orifices (M) open,
respectively.
The kinetic parameters $R_{ev}(X)$ and $k_c(X)$ are calculated from Eqs. (7) and (8) as follows:
$\quad k_c(X) = \frac{k_{esc}^M(X) \cdot [X]_{SS}^M - k_{esc}^S(X) \cdot [X]_{SS}^S}{[X]_{SS}^S - [X]_{SS}^M} - k_w(X) \cdot (1 - \theta)$ $\qquad$ (7)
$\quad R_{ev}(X) = \frac{(k_{esc}^M(X) - k_{esc}^S(X)) \cdot [X]_{SS}^S \cdot [X]_{SS}^M}{[X]_{SS}^S - [X]_{SS}^M} - \frac{N_{TOT}}{V} \cdot k_{des,w}(X) \cdot \theta$ $\qquad$ (8)
This method leads to larger uncertainties for both $R_{ev}(X)$ and $k_c(X)$ compared to the combined
PV and steady state method used before. The reason lies in the fact that two similarly large
numbers, namely $[X]_{SS}^S$ and $[X]_{SS}^M$, are subtracted in the denominators of equations Eqs. (7)
and (8) leading to an uncertain value of $k_c(X)$ and $R_{ev}(X)$. In other words, the noise in the
signal from the MS is such that the two data sets for the small orifice and both orifices open
are sometimes insufficiently linearly independent of each other within experimental
uncertainty.
We also used the combination of real-time PV and steady state experiments using HCl as a
probe gas and applied the experimental method described previously in order to measure the
kinetics of HCl, $R_{ev}(HCl)$ and $\alpha(HCl)$, in the presence of $\alpha$-NAT and $\beta$-NAT ice films.
Once the kinetics $R_{ev}(X)$ and $k_c(X)$ have been measured using the combination of PV and
steady state experiments ($H_2O$, HCl) or the two-orifice method ($HNO_3$, $H_2O$), we may
calculate the equilibrium vapour pressure $P_{eq}(X)$ for each gas according to Eq. (9):
$\quad P_{eq}(X) = \frac{R_{ev}(X)}{k_c(X)} \cdot \frac{RT}{N_A}$ $\qquad$ (9)
where R is the molar gas constant in $cm^3$ Torr $K^{-1}$ $mol^{-1}$, T the temperature of the thin film in
K and $N_A$ Avogadro's constant in molec $mol^{-1}$.



## 3 Results

### 3.1 Crystalline α-NAT Thin Films

The kinetic results for the heterogeneous interaction of $H_2O$ and $HNO_3$ with α-NAT and NAD
thin films obtained in PV and TO experiments are displayed in Figure 2. Full symbols
represent PV experiments: full red circles correspond to experiments on α-NAT substrates,
and full green squares to experiments on NAD substrates. Empty symbols represent TO
experiments with red circles representing $H_2O$ and black triangles $HNO_3$ results. Pure ice
experiments are displayed as inverse blue triangles for comparison purposes. The calculated
relative error for PV experiments is 30% whereas for TO experiments we estimate a relative
error of 60%.
Figure 2a shows the measured accommodation coefficients $\alpha_{\alpha-NAT}(X)$, (X = $H_2O$, $HNO_3$), as
a function of temperature. $\alpha_{\alpha-NAT}(H_2O)$ in PV experiments (full red circles) decreases as a
function of temperature in the range 167-188.5 K, varying from 0.08 at 167 K to $3.1 \times 10^{-3}$ at
188.5 K, which is a factor of 30 lower than $\alpha_{ice}(H_2O)$ on pure ice at the same temperature.
The scatter in the data is not an artifact and is due to the sample-to-sample variability of the
crystalline samples we use and the randomness of the crystalline nucleation process. The
variability may be in surface composition, morphology and smoothness as shown in previous
studies (McNeill et al., 2007; Iannarelli and Rossi, 2014).
$\alpha_{\alpha-NAT}(H_2O)$ in TO experiments (empty red circles) yields different results. For temperatures
lower than 185 K it is equal to $\alpha_{\alpha-NAT}(H_2O)$ on α-NAT in PV experiments within
experimental error. For temperatures higher than 185 K $\alpha_{\alpha-NAT}(H_2O)$ increases as a function
of temperature in contrast to results of PV experiments (full red circles) varying from $8 \times 10^{-3}$
at 183 K to 0.08 at 193.5 K, being equal to $\alpha_{ice}(H_2O)$ on pure ice within experimental error at
the highest temperature. This result compares favorably with the results of Delval and Rossi
(2005) which showed a positive temperature dependence of $\alpha_{\alpha-NAT}(H_2O)$ in the temperature
range 182-207 K. $\alpha_{NAD}(H_2O)$ in PV experiments (green full squares) is equal within
experimental error to $\alpha_{\alpha-NAT}(H_2O)$.
$\alpha_{\alpha-NAT}(HNO_3)$ (black empty triangles) increases as a function of temperature in the measured
temperature range from a value of approximately 0.005 at 181 K to a value of 0.13 at 188 K.
The narrow temperature range follows from the high uncertainties of the two-orifice method
at low temperatures and the increasingly rapid conversion of α-NAT to β-NAT at high





temperatures. These values are lower by a factor of 2 to 40 compared to the preferred values
indicated by the IUPAC Subcommittee on Gas Kinetic Data Evaluation (Crowley et al.,

364    2010).

Figure 2b shows results for the rate of evaporation $R_{ev}(X)$ in molec s$^{-1}$ cm$^{-3}$ as a function of
temperature. The same symbols as for panel (a) are used. $R_{ev}(H_2O)$ on α-NAT in PV
experiments is lower by a factor of 2 compared to $R_{ev}(H_2O)$ on pure ice at temperatures lower
than 175 K. For temperatures higher than 175 K, $R_{ev}(H_2O)$ on α-NAT is lower on average by
up to a factor of 50 compared to $R_{ev}(H_2O)$ on pure ice. This result is very different compared
to the case of HCl where the evaporation of $H_2O$ takes place at a rate characteristic of pure ice
despite the presence of adsorbed HCl on the ice and is in agreement with the findings of
Delval and Rossi (2005).
$R_{ev}(H_2O)$ on α-NAT measured using the TO method is equal within experimental error to
$R_{ev}(H_2O)$ obtained in PV experiments. $R_{ev}(H_2O)$ on NAD is equal to within experimental
error to $R_{ev}(H_2O)$ on α-NAT. The full black line shows the rate of evaporation of pure water
for the system in use, calculated from literature results of the equilibrium vapor pressure
(Marti and Mauersberger, 1993) using α = 1, whereas the dashed black line represents
extrapolated values of $R_{ev}(H_2O)$ for temperatures lower than 173 K using the expression
provided by Mauersberger and coworkers (Marti and Mauersberger, 1993; Mauersberger and
Krankowsky, 2003).
Figure 2c shows the results for $P_{eq}(X)$ in Torr calculated according to Eq. (9) for both $H_2O$
and $HNO_3$ as a function of temperature. The same symbols as in panels (a) and (b) are used.
$P_{eq}(H_2O)$ of α-NAT calculated from the kinetic parameters measured in PV experiments is
lower by a factor of approximately 3 compared to $P_{eq}(H_2O)$ on pure ice at temperatures higher
than 180 K. For temperatures lower than 180 K $P_{eq}(H_2O)$ of α-NAT is close to $P_{eq}(H_2O)$ of
pure ice because the present samples are water-rich (Molina, 1994) with a $HNO_3$ mole
fraction of less than 10%.
$P_{eq}(H_2O)$ of α-NAT calculated from the results of TO experiments is lower by up to a factor
of 10 compared to $P_{eq}(H_2O)$ of pure ice in the temperature range 180-193.5 K. At
temperatures lower than 180 K, $P_{eq}(H_2O)$ of α-NAT from TO experiments is equal within
experimental error to $P_{eq}(H_2O)$ of α-NAT in PV experiments. $P_{eq}(HNO_3)$ of α-NAT is lower
by a factor of 1000 in the temperature range 181-188 K compared to $P_{eq}(H_2O)$ on pure ice.





The values obtained for the equilibrium vapor pressure have been compared with the
HNO$_3$/H$_2$O phase diagram constructed by McElroy et al. (1986); Hamill et al. (1988); Molina
(1994). Figure 3 shows the results for α-NAT and metastable NAD films, PV and TO
experiments. The solid lines represent the coexistence conditions for two phases and the
dashed lines represent vapor pressures of liquids with composition given as % (w/w) of
HNO$_3$. The shaded rectangular area represents typical polar stratospheric conditions. The
slope m of the coexistence lines depends on the difference of the enthalpies of sublimation of
the two acid hydrate species, namely NAM and NAT, according to Eq. (10) (Wooldridge et
al., 1995):
$$m = \frac{\Delta H^1_{subl} - \Delta H^2_{subl}}{(n_1 - n_2)\, R} \qquad\qquad (10)$$
where $\Delta H^1_{subl}$ and $\Delta H^2_{subl}$ are the enthalpies of sublimation of the acid hydrates in kJ/mol, $n_1$
and $n_2$ the number of water molecules of the respective hydrate and R is the gas constant in J
mol$^{-1}$ K$^{-1}$. The slope of the ice/NAT coexistence line is calculated from Wooldridge et al.
(1995) as $m_{ice/NAT}$ = (50.9 kJ/mol)/R and the slope of the NAT/NAM coexistence line is
calculated as $m_{NAT/NAM}$ = (55.9 kJ/mol)/R.
All α-NAT experiments lie in the existence area of nitric acid trihydrate, as expected. On the
other hand, α-NAT under polar stratospheric conditions (shaded rectangular area) is unstable
and starts to convert into the stable β-NAT phase (Koehler et al., 1992). The small number of
α-NAT samples we reported in the shaded gray area is further confirmation of results reported
in the literature. NAD samples are expected to lie closer to the monohydrate region, given
their composition close to the H$_2$O:HNO$_3$ = 2:1 stoichiometry (Iannarelli and Rossi, 2015).
Nevertheless, the pure ice phase is still dominant in our samples and all our samples are
water-rich (Molina, 1994) with a HNO$_3$ mole fraction, even in NAD films, of less than 10%.
**3.2   Crystalline β-NAT Thin Films**
The results for β-NAT thin films obtained in PV and TO experiments are displayed in Figure
4. Full and empty red squares represent PV and TO experiments, respectively, with red
squares representing H$_2$O and black triangles HNO$_3$ results. Pure ice experiments are
displayed as inverse blue triangles for comparison. The calculated relative error for PV
experiments is 30% whereas for TO experiments we estimate a relative error of 60%.


Figure 4a shows the measured $\alpha_{\beta-NAT}(X)$ as a function of temperature. $\alpha_{\beta-NAT}(H_2O)$ in PV
experiments (full red squares) shows scatter similar to the case of $\alpha_{HH}(HCl)$ on crystalline
HCl hexahydrate (Iannarelli and Rossi, 2014). Also in this case, a variation of up to a factor of
10 for results at the same temperature is observed. We may interpret this result like in the HCl
hexahydrate case where the scatter may be caused by the variability of the surface
composition, the morphology or the smoothness of the ice surface (McNeill et al., 2007).
Similar results have recently been presented by Moussa et al. (2013) regarding the nitric acid-
induced surface disorder on ice. In any case, all results show that $\alpha_{\beta-NAT}(H_2O)$ is at least a
factor of 10 lower than $\alpha_{ice}(H_2O)$ on pure ice in the temperature range 182-200 K.
$\alpha_{\beta-NAT}(H_2O)$ in TO experiments (empty red squares) on the other hand, increases as a
function of temperature in the temperature range 182-198 K varying from 0.013 at 182 K to
approximately 0.1 at 198 K, being equal at the highest temperature to $\alpha_{ice}(H_2O)$ on pure ice
within experimental error. This result is in contrast to Delval and Rossi (2005) who report a
negative temperature dependence of $\alpha_{\beta-NAT}(H_2O)$ in the temperature range 182-207 K. A
possible reasons for the different behavior of PV and TO experiments may be intrinsic in the
nature of PV experiments: the ice surface is exposed to a series of pulses of $H_2O$ and the free
sites may be saturated before the introduction of each consecutive pulse. We suspect this may
be the reason for the discrepancy between PV and TO experiments and we will consider the
results of TO experiments as the preferred values of this work despite the larger experimental
scatter.
Like $\alpha_{\beta-NAT}(H_2O)$, the values of $\alpha_{\beta-NAT}(HNO_3)$ (black empty triangles) increase as a
function of temperature in the measured temperature range from a value of approximately
0.015 at 182 K to a value of 0.08 at 195.5 K. However, the values have a large estimated
uncertainty. These values are lower by a factor of 2 to 10 compared to the preferred values
indicated by the IUPAC Subcommittee on Gas Kinetic Data Evaluation (Crowley et al., 2010)
in the temperature range 190 to 200 K.
Figure 4b shows results for $R_{ev}(X)$ in molec $s^{-1}$ $cm^{-3}$ as a function of temperature. The same
symbols as in panel (a) are used. $R_{ev}(H_2O)$ on $\beta$-NAT in PV experiments is lower by a factor
of 50 compared to $R_{ev}(H_2O)$ on pure ice in the temperature range 182-200 K. As in the case of
$\alpha$-NAT, this result is very different compared to the case of HCl where the evaporation of
$H_2O$ is not influenced by the presence of adsorbed HCl on the ice and takes place at a rate
characteristic of pure ice for all HCl concentrations used.



$R_{ev}(H_2O)$ on β-NAT measured using the TO method is close to $R_{ev}(H_2O)$ obtained in PV
experiments, the former being approximately a factor of 2 higher. $R_{ev}(HNO_3)$ on β-NAT
increases in the temperature range 182-195.5 K with a steeper slope compared to $R_{ev}(H_2O)$,
the former being smaller by approximately a factor of 1000 at low and 50 at higher
temperature compared to $R_{ev}(H_2O)$ of β-NAT. It varies from $2 \times 10^8$ at 182 K to $8.5 \times 10^9$ molec
$s^{-1}$ $cm^{-3}$ at 195.5 K.
Figure 4c shows the results for $P_{eq}(X)$ in Torr calculated according to Eq. (9) for both $H_2O$
and $HNO_3$ as a function of temperature. The same symbols as in panels (a) and (b) are used.
$P_{eq}(H_2O)$ of β-NAT calculated from the results of TO experiments is lower by up to a factor
of 10 in the middle of the covered T-range compared to $P_{eq}(H_2O)$ of pure ice in the
temperature range 182-195.5 K. $P_{eq}(H_2O)$ of β-NAT calculated from the kinetic parameters
measured in PV agrees with TO experiments within experimental uncertainty. Saturation
effects in PV experiments will affect both the accommodation (α) and evaporation ($J_{ev}$)
process to the same extent such that $P_{eq}$ should be invariant to the chosen experimental
procedure (PV or TO).
The scatter of $P_{eq}(H_2O)$ is of the same magnitude as the scatter of $\alpha_{\beta-NAT}(H_2O)$ and may
likewise be explained by an increase in the substrate roughness or inhomogeneous nature of
the β-NAT surface owing to exposure to repetitive transient saturation of $H_2O$ in the
aftermath of each pulse.
Figure 5 shows the $HNO_3/H_2O$ phase diagram with the results obtained for β-NAT films: all
β-NAT experiments lie in the existence area of nitric acid trihydrate and the majority of points
are in the rectangular shaded area representing polar stratospheric conditions. As already
mentioned, β-NAT is the stable phase under these conditions and our results agree well with
the literature (McElroy et al., 1986; Hamill et al., 1988; Molina, 1994; Koehler et al., 1992).
**3.3  HCl kinetics on α-NAT and β-NAT Thin Films**
As already mentioned, we used a combination of real-time PV and steady state experiments
using HCl as probing gas in order to measure the kinetics of HCl interacting with  α-NAT and
β-NAT ice films.
The current experimental setup does not allow the measurement of the kinetics of 3 gases at
the same time. We therefore had to make some assumptions and/or simplifications in order to



measure the unknown parameters of Eq. (2) for each gas used. Specifically, we made the
following assumptions, both for α-NAT and β-NAT substrates:

486       • $R_{ev}(H_2O)$ on NAT remains unchanged in the presence of HCl

487       • $\alpha_{NAT}(H_2O)$ remains unchanged in the presence of HCl

488       • $\alpha_{NAT}(HNO_3)$ remains unchanged in the presence of HCl

Under these assumptions, no additional measurements of the heterogeneous kinetics of $H_2O$
in the presence of HCl have been performed. We have measured the steady-state flow
$F_{SS}(HNO_3)$ before each HCl pulse series and used previously measured $\alpha_{\alpha-NAT}(HNO_3)$ and
$\alpha_{\beta-NAT}(HNO_3)$ from TO experiments on α-NAT and β-NAT phases in order to calculate
$R_{ev}(HNO_3)$ and $P_{eq}(HNO_3)$ according to Eqs. (8) and (9) in HCl-PV experiments as well. As a
net result we measure or calculate the following kinetic parameters for α-NAT and β-NAT
substrates: $R_{ev}(HCl)$, $\alpha_{NAT}(HCl)$ and $R_{ev}(HNO_3)$ in the presence of HCl.
Figure 6 displays the results of HCl-PV experiments on α-NAT substrates. Full red diamonds
represent the results for HCl whereas full black circles represent $HNO_3$ results using
$\alpha_{\alpha-NAT}(HNO_3)$ from TO experiments and $F_{SS}(HNO_3)$ from HCl-PV experiments. Empty
black triangles represent results for $HNO_3$ in TO experiments reported from Figure 2 for
comparison.
Figure 6a displays the measured $\alpha_{\alpha-NAT}(X)$ as a function of temperature. $\alpha_{\alpha-NAT}(HCl)$ (full
red diamonds) slightly decreases as a function of temperature in the range 177.5-199.5 K,
being equal to $\alpha_{ice}(H_2O)$ on pure ice at low temperatures and lower by a factor of 4 at T =
199.5 K. Values of $\alpha_{\alpha-NAT}(HNO_3)$ measured in TO experiments in the absence of HCl are
reported as empty black triangles. We used these values in order to calculate $R_{ev}(HNO_3)$ and
$P_{eq}(HNO_3)$ in the presence of HCl.
Figure 6b shows results for $R_{ev}(X)$ in molec s$^{-1}$ cm$^{-3}$ as a function of temperature. The same
symbols as in panel (a) are used. $R_{ev}(HCl)$ on α-NAT slightly increases as a function of
temperature and is lower by a factor of 1000 in the measured temperature range 177.5-199.5
K compared to $R_{ev}(H_2O)$ on pure ice. $R_{ev}(HNO_3)$ increases as a function of temperature,
varying from $1\times10^8$ at 181 K to $9\times10^9$ molec s$^{-1}$ cm$^{-3}$ at 189 K. The presence of HCl does not
have any effect on the rate of evaporation of $HNO_3$ from α-NAT films: we observe no
increase of $F_{ss}(HNO_3)$ following HCl pulses and $R_{ev}(HNO_3)$ in the presence of adsorbed HCl
molecules (full black circles) is identical within experimental error to $R_{ev}(HNO_3)$ of α-NAT




films free of adsorbed HCl (empty black triangles). However, this result is contingent upon
the assumptions listed before, namely $\alpha_{\alpha-NAT}(HNO_3)$ being independent of the presence or
absence of HCl.
Figure 6c shows the results for $P_{eq}(X)$ in Torr calculated according to Eq. (9) for both HCl and
$HNO_3$ as a function of temperature. The same symbols as in panel (a) and (b) are used.
$P_{eq}(HCl)$ of α-NAT is lower by a factor of approximately 100 compared to $P_{eq}(H_2O)$ on pure
ice in the measured temperature range. A comparison with the results of $P_{eq}(HCl)$ of
crystalline HCl hexahydrate and amorphous $HCl/H_2O$ mixtures calculated using the same
experimental methodology (Iannarelli and Rossi, 2014) shows that $P_{eq}(HCl)$ of α-NAT is
lower by a factor of approximately 10 compare to $P_{eq}(HCl)$ of crystalline hexahydrate in the
overlapping temperature range (177.5-193.5 K).
$P_{eq}(HCl)$ of amorphous $HCl/H_2O$ mixtures is higher by a factor of 20 compared to $P_{eq}(HCl)$ of
α-NAT at low temperatures (177.5 K) with the difference decreasing at high temperatures
(199.5 K) where $P_{eq}(HCl)$ of the amorphous mixture is only a factor of 4 higher than $P_{eq}(HCl)$
of α-NAT.
$P_{eq}(HNO_3)$ on HCl-doped α-NAT films is equal within experimental error to $P_{eq}(HNO_3)$ of α-
NAT films free of adsorbed HCl. It is lower by a factor of 1000 compared to $P_{eq}(H_2O)$ on
pure ice in the measured temperature range 177.5-199.5 K.
Figure 7a (symbols have the same meaning as in Figure 6) shows the measured values of
$\alpha_{\beta-NAT}(X)$ as a function of temperature. $\alpha_{\beta-NAT}(HCl)$ slightly decreases as a function of
temperature in the range 177-201 K, varying from 0.025 at 177 K to 0.016 at 201 K. As for
the case of α-NAT, we assume that $\alpha_{\beta-NAT}(HNO_3)$ (empty black triangles) equals the
measured values of $\alpha_{\beta-NAT}(HNO_3)$ on HCl-free β-NAT in two-orifice experiments whose
results are displayed in Figure 4a.
Figure 7b shows results for the $R_{ev}(X)$ in molec $s^{-1}$ $cm^{-3}$ as a function of temperature. The
same symbols as in Panel (a) are used. $R_{ev}(HCl)$ on β-NAT is equal at higher temperature
within experimental uncertainty to $R_{ev}(HCl)$ on α-NAT and is lower by a factor of 1000 in the
temperature range 177- 201 K compared to $R_{ev}(H_2O)$ on pure ice. $R_{ev}(HNO_3)$ on HCl-doped
β-NAT films, being equal within experimental error to $R_{ev}(HNO_3)$ of undoped β-NAT films,
indicates that adsorbed HCl molecules seem to have no effect on the rate of evaporation of
$HNO_3$ from β-NAT films in the presence of HCl as well.





Figure 7c shows the results for $P_{eq}(X)$ in Torr calculated according to Eq. (9) for both HCl and
$HNO_3$ as a function of temperature. The same symbols as in panel (a) and (b) are used.
$P_{eq}(HCl)$ of $\beta$-NAT is lower by a factor of approximately 100 compared to $P_{eq}(H_2O)$ on pure
ice. $P_{eq}(HCl)$ of $\beta$-NAT is identical within experimental uncertainty to $P_{eq}(HCl)$ of $\alpha$-NAT in
the measured temperature range 177-201 K and the same observations are valid when
comparing $P_{eq}(HCl)$ of crystalline HCl hexahydrate with amorphous $HCl/H_2O$ mixtures
(Iannarelli and Rossi, 2014).

**4   Discussion**
In this work we have been able to grow $HNO_3$ hydrates at temperatures relevant to the
stratosphere with tight control on the deposition conditions whose details have been published
by Iannarelli and Rossi (2015) as far as the mass balance is concerned. Direct crystallization
of $\alpha$-NAT film on pure ice has been observed upon $HNO_3$ deposition. Under the present
system conditions $\beta$-NAT was never observed to crystallize directly upon $HNO_3$ deposition
but was always obtained as the stable form after conversion of $\alpha$-NAT films. Temperatures
higher than 185 K are necessary for the conversion to occur on the time scale of the
experiments we have performed.
$\alpha_{\alpha-NAT}(H_2O)$ shows two distinct temperature dependent regimes. At temperatures lower than
180-185 K it decreases as a function of temperature reaching a minimum of approximately
0.003 at 185 K as displayed in Figure 2a. For temperatures higher than 185 K, $\alpha_{\alpha-NAT}(H_2O)$
increases as a function of temperature, being equal to $\alpha_{ice}(H_2O)$ on pure ice and $\alpha_{\beta-NAT}(H_2O)$
at 193.5 K. An Arrhenius representation of the evaporative flux $J_{ev}(H_2O)$ (see Table 1) on $\alpha$-
NAT shows two distinct regimes of temperature dependence, as well. Figure 8 reports the
results for PV and TO experiments as full and empty red circles, respectively. We keep the
two data sets separated for clarity, but the results of PV and TO experiments are
indistinguishable within experimental uncertainty in the measured temperature range.
Eqs. (11) and (12) present the two-parameter representations of the Arrhenius lines for
$J_{ev}(H_2O)$ displayed in Figure 8. Equations (11) and (12) represent the solid and dashed red
lines, respectively, with R = 8.314 J $K^{-1}$ $mol^{-1}$ used throughout:
181 K $\leq$ T $\leq$ 193.5 K: $\log J_{ev}(H_2O)[molec \cdot cm^{-2} \cdot s^{-1}] = (35.9 \pm 2.8) - \frac{(75.3 \pm 9.9) \times 10^3}{2.303 \, RT}$

576             (11)





$167\ \text{K} \leq T \leq 181\ \text{K}: \log J_{ev}(H_2O)[\text{molec} \cdot \text{cm}^{-2} \cdot \text{s}^{-1}] = (15.1 \pm 1.2) - \frac{(3.5 \pm 4.2) \times 10^3}{2.303\ RT}$ (12)
Table 3 reports a synopsis of the kinetic ($J_{ev}$) as well as the thermodynamic ($P_{eq}$) parameters
calculated for all experiments of the present work.
The considerable scatter in the data, reflected in the significant uncertainties of Eqs. (11) and
(12), may be explained by the variability of the surface composition of the film as well as the
surface roughness and surface disorder of the ice substrates, in analogy to the HCl case
(Iannarelli and Rossi, 2014). For HCl the scatter in the kinetic data was thought to be due to
the stochastic nature of crystal growth of hexahydrate films compared to amorphous mixtures
of HCl/$H_2O$ of similar composition and does not represent a lack of reproducibility.
Moussa et al. (2013) have observed variations of up to a factor of 10 of the $HNO_3$ vapor
pressure of "smooth" ice samples exposed to $HNO_3$ as a result of induced surface disorder.
The exposure of the present samples to repeated high $H_2O$ supersaturation during PV
experiments may lead to surface increased disorder due to liquefaction and/or reconstruction.
In the high temperature regime we calculate an activation energy for $H_2O$ evaporation
$E_{ev}(H_2O) = (75.3 \pm 9.9)\ \text{kJ mol}^{-1}$, and in the low temperature regimes almost no temperature
dependence is observed with an activation energy for $H_2O$ evaporation of $E_{ev}(H_2O) = (3.5 \pm$
$4.2)\ \text{kJ mol}^{-1}$.
The discontinuity in the Arrhenius representation of kinetic parameters has already been
observed in pure ice as reported by Chaix et al. (1998); Delval et al. (2003); Delval and Rossi
(2004); Pratte et al. (2006). The temperatures at which the discontinuity occurs are higher in
previous work: Delval et al. (2003) reported a discontinuity at approximately 208 K in their
work on $H_2O$ evaporation from HCl and HBr doped ice substrates. In a quartz crystal
microbalance study of $H_2O$ evaporation from pure ice the change in slope is reported at $193 \pm$
$2$ K (Delval and Rossi, 2004) comparable with the temperature of $188 \pm 2$ K reported by
Pratte et al. (2006) in their work on the kinetics of $H_2O$ evaporation and condensation on
different types of ice.
No clear explanation for this break has yet been advanced. The discontinuity may be an
indication of the formation of a new disordered structure similar to the quasi-liquid layer
induced by HCl as proposed by McNeill et al. (2006). The observation of the break in pure ice
samples as well, however, strongly suggests that the onset of a quasi-liquid layer may be
independent of the presence of HCl and that the history and evolution of the sample play a
role in the arrangement of the structure, similarly to the case of the presence of cubic ice at





high temperature in common hexagonal ice that finally turned out to be a perturbed hexagonal
ice structure (Kuhs et al., 2012).
In the case of β-NAT we have good agreement between PV (dotted line) and TO (solid line)
experiments of $P_{eq}(H_2O)$ as shown in the van 't Hoff representation displayed in Figure 9.
As already mentioned, the ice surface is exposed to a series of pulses of $H_2O$ during PV
experiments. The free sites may be saturated before the introduction of each consecutive pulse
resulting in the discrepancy between PV and TO experiments. We therefore believe that the
results from PV experiments are more precise but less accurate owing to partial surface
saturation whereas the TO experiments are less precise but more accurate. We chose the latter
as the preferred values of this work despite the larger scatter in the data compared to the PV
experiments.
Eqs. (13) and (14) reports the best linear fit for TO and PV experiments displayed in Figure 9,
respectively:
$$\log P_{ev}(H_2O)[\text{Torr}] = (16.7 \ \pm \ 4.9) - \frac{(76.7 \pm 17.7) \times 10^3}{2.303 \ RT} \qquad \text{TO – Preferred} \qquad (13)$$

$$\log P_{ev}(H_2O)[\text{Torr}] = (16.7 \ \pm \ 3.0) - \frac{(75.5 \pm 11.1) \times 10^3}{2.303 \ RT} \qquad \text{PV} \qquad (14)$$

The enthalpies of evaporation of $H_2O$ on β-NAT films calculated for the two measurement
techniques are $\Delta H^0_{ev,TO}(H_2O) = (76.7 \pm 17.7)$ kJ mol$^{-1}$ for TO and $\Delta H^0_{ev,PV}(H_2O) = (75.5 \pm$
11.1) kJ mol$^{-1}$ for PV experiments, respectively. The results show good agreement between
the two experimental techniques despite the experimental scatter. The average value of
$\Delta H^0_{ev}(H_2O) = (76.1 \pm 14.4)$ kJ mol$^{-1}$ is slightly higher, as expected, but not significantly
different compared to α-NAT films. Figure S2 of Supplement C displays a van't Hoff plot for
α-NAT with $\Delta H^0_{ev}(H_2O) = (70.3 \pm 14.1)$ and $(56 \pm 5.1)$ kJ mol$^{-1}$ for TO and PV experiments,
respectively. Both values are identical within experimental uncertainty whose average yields
$\Delta H^0_{ev}(H_2O) = (63.4 \pm 9.6)$ kJ mol$^{-1}$ and which leads to a standard enthalpy of formation
slightly larger than that for β-NAT, as expected.
However, we do not have good agreement between TO and PV experiments for the kinetic
parameters of β-NAT: a discrepancy is observed in the results of the two measurement
techniques regarding $R_{ev}(H_2O)$ and $\alpha(H_2O)$ for β-NAT. Figure 4 already shows a discrepancy
in $\alpha(H_2O)$ (full and empty red squares in panel a) with the results of TO experiments being
larger by a factor of approximately 5 at 185 K increasing to a factor of 100 at 200 K compared





to PV experimental results across the whole temperature range. The same qualitative trend,
albeit to a smaller extent, is observed for $R_{ev}(H_2O)$ (Figure 4b) and the Arrhenius
representation of $J_{ev}(H_2O)$ on β-NAT clearly shows the discrepancy between the different
measurement techniques.
The two-parameter representations of the Arrhenius lines displayed in Figure 10 are reported
in Eqs. (15) and (16) for TO (solid line) and PV (dotted line) experiments, respectively:
$\log J_{ev}(H_2O)[molec \cdot cm^{-2} \cdot s^{-1}] = (36.0 \pm 1.3) - \frac{(77.0 \pm 4.9) \times 10^3}{2.303\,RT}$  TO – Preferred     (15)
$\log J_{ev}(H_2O)[molec \cdot cm^{-2} \cdot s^{-1}] = (28.7 \pm 0.7) - \frac{(52.1 \pm 2.4) \times 10^3}{2.303\,RT}$  PV                (16)
Contrary to the case of α-NAT, no discontinuity in $J_{ev}(H_2O)$ has been observed in the
Arrhenius plot of β-NAT displayed in Figure 10. We attribute the discrepancy between PV
and TO experiments to the fact that the former may be subject to partial saturation of uptake
and evaporation in the aftermath of transient supersaturation (PV). A look at the results of
$\alpha_{\alpha-NAT}(H_2O)$ in Figure 2a reveals that the results of the TO measurement technique agrees
well with the PV technique in the overlapping temperature range. However, this plot displays
a "hole" of a factor of 20 centered around T = 180 ± 3 K with respect to the values at the
fringes of the temperature interval. There are indications that PV experiments on α-NAT
substrates may yield lower values of $\alpha_{\alpha-NAT}(H_2O)$ at high temperatures in excess of
approximately 182 K (Figure 2a), similarly to the results for $\alpha_{\beta-NAT}(H_2O)$ for a β-NAT film
(Figure 4a). This might be an indication that PV experiments are very sensitive to the
interfacial nature of the sample. In other words, transient supersaturation (PV) and "passive"
steady-state (TO) experiments may address different properties of the gas-condensed surface
interface. This is the first time such a large discrepancy between two kinetic measurements
techniques has been observed.  As expected, thermodynamic results are not affected for
reasons of microscopic reversibility because both forward $(\alpha(H_2O))$ and reverse reactions
$(J_{ev}(H_2O))$ are affected to the same extent which cancels out for the calculation of the values
of thermodynamic parameters.
Figure S3 of Supplement C shows the results of PV experiments using $H_2O$ as a probe gas on
α-NAT and β-NAT substrates. Red and black circles represent the decay of series of two
pulses on α- and β-NAT, respectively, with the first and second pulse labeled accordingly. In
the case of α-NAT films (red circles), the decay of the second pulses is equal to within 20-



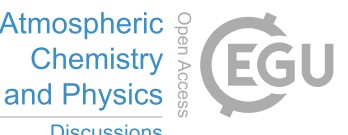

30% of the decay of the initial pulses, and only in a few cases at temperatures higher than 180
K is the decay of the second pulse significantly slower than the initial pulse. In the case of β-
NAT films, the decay of second pulses is consistently slower than the decay of first pulses in
most cases. This indicates that the surface of β-NAT films exposed to a transient
supersaturation of $H_2O$ vapor is more prone to saturation compared to α-NAT.
As mentioned before, we consider the results of TO experiments as preferred for this work
despite the larger uncertainty. The enthalpies of evaporation $\Delta H^0_{ev,TO}(H_2O) = (76.7 \pm 17.7)$ kJ
$mol^{-1}$ and the activation energy for evaporation $E_{ev}(H_2O) = (77.0 \pm 4.9)$ kJ $mol^{-1}$ are equal to
within experimental uncertainties. We calculate an activation energy of accommodation for
$H_2O$ on β-NAT of $E_{acc}(H_2O) = E_{ev}(H_2O) - \Delta H^0_{ev,TO}(H_2O) = 0$. Therefore, no activation energy
is required for the accommodation process of $H_2O$ on β-NAT which is an expected
experimental outcome. In contrast, the activation energy for $H_2O$ accommodation on α-NAT
is computed as $E_{acc}(H_2O) = E_{ev}(H_2O) - \Delta H^0_{ev,average}(H_2O) = 75.3 - 63.4 = 11.9$ kJ/mol when
using a value averaged over the PV and TO experiment of 63.4 kJ/mol for $\Delta H^0_{ev,average}(H_2O)$.
This small, but possibly significant positive activation energy is consistent with the
temperature dependence of $\alpha_{\beta-NAT}(H_2O)$ displayed in Figure 4a for the TO experiment.
$R_{ev}(H_2O)$ on both α-NAT and β-NAT is smaller compared to $R_{ev}(H_2O)$ on pure ice. This is in
agreement with the results of Tolbert and Middlebrook (1990) and Delval and Rossi (2005)
who showed that ice coated with a layer of NAT evaporates at a slower rate than pure ice. On
the other hand, our results are in contrast with the findings of Biermann et al. (1998) who
report that no significant decrease of the $H_2O$ evaporation rate was observed in $HNO_3$-doped
ice films. The discrepancy may be due to the low $HNO_3$ concentration used by Biermann et
al. (1998) compared to our experimental conditions as well as probable wall losses due to
$HNO_3$-wall interaction which was not taken into account in contrast to the present approach .
Delval and Rossi (2005) report that the initial evaporation of $H_2O$ in their experiments was
always that of pure ice and that $R_{ev}(H_2O)$ gradually decreases with the evaporation of excess
$H_2O$ and the increase in the average $HNO_3$ mole fraction. They refer to this difference as
"high and low evaporation rate" regime of $H_2O$.
Our observation is somewhat different: $R_{ev}(H_2O)$ on α-NAT and β-NAT is smaller compared
to $R_{ev}(H_2O)$ on pure ice over the whole temperature range and for all samples. The reason lies
in the fact that the average mole fraction of $HNO_3$ of the present samples is higher by at least





a factor of 10 compared to the one used by Delval and Rossi (2005). Therefore all our
samples are in the "low evaporation rate" regime of $H_2O$ and our results compare well with
the results of Delval and Rossi (2005) once they evaporate excess $H_2O$ and reach the "low
evaporation rate" regime.
Figure 11 displays both the Arrhenius plots of $J_{ev}(HNO_3)$ (A) and the van 't Hoff plots of
$P_{eq}(HNO_3)$ (B) for the interaction of $HNO_3$ with $\alpha$- and $\beta$-NAT films. We would like to
briefly remind the reader that only TO experiments were possible for $HNO_3$ experiments. The
following equations define the corresponding straight lines based on the present
measurements. For $\alpha$-NAT (Eqs. (17) and (18)) and $\beta$-NAT films (Eqs. (19) and (20)) we find
the following results:
$\alpha$-NAT:  $\log J_{ev}(HNO_3)[molec \cdot cm^{-2} \cdot s^{-1}] = (62.3 \pm 7.8) - \frac{(178.0 \pm 27.4)\times 10^3}{2.303\ RT}$  (17)
$\log P_{ev}(HNO_3)[Torr] = (29.3 \pm 12.0) - \frac{(128.6 \pm 42.4)\times 10^3}{2.303\ RT}$  (18)
$\beta$-NAT:  $\log J_{ev}(HNO_3)[molec \cdot cm^{-2} \cdot s^{-1}] = (40.6 \pm 2.4) - \frac{(102.0 \pm 8.6)\times 10^3}{2.303\ RT}$  (19)
$\log P_{ev}(HNO_3)[Torr] = (19.8 \pm 3.3) - \frac{(96.5 \pm 12.0)\times 10^3}{2.303\ RT}$  (20)
We calculate an activation energy for $HNO_3$ evaporation on $\alpha$-NAT and $\beta$-NAT of
$E_{ev}(HNO_3) = (178.0 \pm 27.4)$ kJ mol[-1] and $E_{ev}(HNO_3) = (102.0 \pm 8.6)$ kJ mol[-1], respectively.
These values are higher compared to $E_{ev}(HCl) = (87.0 \pm 17)$ kJ mol[-1], the activation energy
for HCl evaporation on hexahydrate. This result is within expectation given the higher
hydrogen bond energy of $HNO_3$ compared to HCl with $H_2O$.
Similar to the case of $H_2O$, no activation energy for accommodation of $HNO_3$ on $\beta$-NAT is
required since the evaporation activation energy $E_{ev}(HNO_3) = (102.0 \pm 8.6)$ kJ mol[-1] and the
enthalpy of evaporation $\Delta H_{ev}^0(HNO_3) = (96.5 \pm 12.0)$ kJ mol[-1] are equal within experimental
uncertainties. In contrast, a substantial activation energy of $H_2O$ mass accommodation of 49.4
kJ/mol is calculated from $E_{acc}(H_2O) = E_{ev}(H_2O) - \Delta H_{ev,TO}^0(H_2O) = 178.0 - 128.6 = 49.9$
kJ/mol which may have to do with the fact that $\alpha$-NAT is metastable owing to its unstable
$H_2O$ crystal structure.
The thermodynamic parameters obtained above, namely $\Delta H_{ev}^0(H_2O)$ and $\Delta H_{ev}^0(HNO_3)$ for
both $\alpha$- and $\beta$-NAT may now be used to estimate the relative stability of $\alpha$- vs. $\beta$-NAT as
follows. The evaporation/condensation equilibrium for both forms of NAT may be





represented in equation (21) where $\Sigma\Delta H_{ev}^{0} = 3\,\Delta H_{ev}^{0}(H_2O) + \Delta H_{ev}^{0}(HNO_3)$ in agreement with
the relevant stoichiometry:
$HNO_3 \bullet 3H_2O(s) \leftrightarrows 3H_2O(g) + HNO_3(g)$         $(\Sigma\Delta H_{ev}^{0})$             (21)
For α- and β-NAT we obtain $\Sigma\Delta H_{ev}^{0,\alpha}$ and $\Sigma\Delta H_{ev}^{0,\beta}$ equal to 318.8 and 324.8 kJ/mol,
respectively, when we use the average of the TO and PV experiment for $H_2O$ and the TO
value listed above for $HNO_3$ evaporation. Specifically, we have used (63.4 ± 9.6) and (128.6
± 42.2) for $H_2O$- and (76.1 ± 14.4) and (96.5 ± 12.0) for $HNO_3$-evaporation for α- and β-
NAT, respectively, as displayed above. Finally, we arrive at the difference $\Sigma\Delta H_{ev}^{0,\alpha}$ -
$\Sigma\Delta H_{ev}^{0,\beta}$ = -6.0 ± 20.0 kJ/mol which shows that β-NAT is marginally more stable than α-
NAT. This is true despite the fact that the standard heat of evaporation for $HNO_3$ in α-NAT
$(\Delta H_{ev}^{0}(HNO_3))$ is significantly larger than for β-NAT by 32.1 kJ/mol which may be expressed
by the fact that the calculated "affinity" of $HNO_3$ towards ice in the α-NAT is larger than for
β-NAT as claimed by Weiss et al. (2016). However, this fact only addresses the behavior of
$HNO_3$ without taking into consideration the partial stability of the $H_2O$ network in the total
crystal structure. In view of the large uncertainty, mainly brought about by the TO
experiment, we regard this result as an estimate to the true standard enthalpy difference
between α- and β-NAT.
The results of HCl kinetic measurements displayed in Figure 6 and Figure 7 show that
$R_{ev}(HCl)$ is always higher than $R_{ev}(HNO_3)$, with the latter being equal regardless of the
presence of absorbed HCl molecules in the condensed phase. Hynes et al. (2002) also
observed that HCl uptake on $HNO_3$ dosed ice was always nearly reversible in their
experiments, in contrast to HCl uptake on clean ice. Although the same $HNO_3$ dosed ice
surface has been dosed repeatedly at different HCl concentrations by Hynes et al. (2002), the
degree of reversibility was found to be unaffected by previous experiments. In contrast, we
never observed such reversibility. In our experiments, HCl always remained on the surface,
evaporating at a rate only slightly faster than $HNO_3$ both for α-NAT and β-NAT and similarly
to $R_{ev}(HCl)$ of crystalline hexahydrate (Iannarelli and Rossi, 2014). However, a possible
influence of the temperature cannot be excluded at this time, as the experiments performed by
Hynes et al. (2002) have been performed at distinctly higher temperatures, namely in the
range 210-235 K, compared to the experiments discussed here.



Similar behavior has been observed by Kuhs et al. (2012) with respect to the presence of
cubic ice or "ice $I_c$" in common hexagonal ice $I_h$. $I_h$ is expected to be the prevalent ice phase at
temperatures relevant to atmospheric processing on thermodynamic grounds. Apparent
formation of $I_c$ has been observed over a wide temperature range and evidence pointed
towards the fact that the resulting phase is not pure cubic ice but instead composed of
disordered cubic and hexagonal stacking sequences. Kuhs et al. (2012) studied the extent and
relevance of the stacking disorder using both neutron as well as X-ray diffraction as indicators
of the "cubicity" of vapor deposited ice at temperatures from 175 to 240 K and could simply
not find proof for the formation of cubic ice $I_c$ under atmospheric conditions.
Kuhs et al. (2012) discovered that even at temperatures as high as 210 K, the fraction of cubic
sequences in vapor deposited ice is still approximately 40%. The rate of decrease in cubicity
depends on the temperature, being very slow at temperatures lower than 180 K and
increasingly rapid at temperatures higher than 185 K. Furthermore, even at high temperatures
the complete transformation into pure ice $I_h$ was never observed, with a few percent of cubic
stacking sequences still remaining in the ice, even after several hours at 210 K and
disappeared only upon heating to 240 K. In addition, the combination of neutron and X-ray
diffraction experiments of Kuhs et al. (2012) cannot distinguish the difference between the
bulk and the interface whereas our measurement techniques, in particular PV experiments, are
very sensitive to the nature and properties of the sample interface.
In light of these results we speculate that the presence of two hydrates of $HNO_3$, namely α-
NAT and β-NAT, may depend on the cubicity or stack-disorder of the ice upon which the
NAT grows. $HNO_3$ adsorbed on cubic ice $I_c$ tends to form α-NAT crystalline structures which
upon heating converts to β-NAT while the ice loses part of its cubicity. The temperature at
which the conversion from α-NAT to β-NAT is accelerated, T = 185 K, is the same
temperature Kuhs et al. (2012) report as the temperature at which the rate of decrease in
cubicity increases. Our hypothesis is that the formation of α-NAT or β-NAT may highly
depend on the environment in which the NAT phase grows and on the presence of high or low
fractions of "$I_c$".
Figure 12 displays both the Arrhenius plots of $J_{ev}(HCl)$ (A) and the van 't Hoff plots of
$P_{eq}(HCl)$ (B) for the interaction of HCl with α-NAT and β-NAT films. As for the case of
$HNO_3$, only TO experiments were performed with HCl as a probe gas. Full red circles and
black triangles represent the interaction of HCl with α- and β-NAT films, respectively.



The following equations define the corresponding straight lines resulting from the present
measurements. For α-NAT (Eqs. (22) and (23)) and β-NAT films (Eqs. (24) and (25)) we find
the following results:
α-NAT: $\quad \log J_{ev}(HCl)[molec \cdot cm^{-2} \cdot s^{-1}] = (34.8 \pm 5.3) - \frac{(78.3 \pm 19.2) \times 10^3}{2.303\ RT}$      (22)
$\log P_{ev}(HCl)[Torr] = (15.7 \pm 3.2) - \frac{(78.4 \pm 11.4) \times 10^3}{2.303\ RT}$      (23)
β-NAT: $\quad \log J_{ev}(HCl)[molec \cdot cm^{-2} \cdot s^{-1}] = (28.6 \pm 1.3) - \frac{(56.7 \pm 4.6) \times 10^3}{2.303\ RT}$      (24)
$\log P_{ev}(HCl)[Torr] = (13.3 \pm 1.6) - \frac{(69.6 \pm 5.8) \times 10^3}{2.303\ RT}$      (25)
Despite the scatter of the data displayed in Figure 12 it may be pointed out that the enthalpy
of HCl evaporation is identical for α- and β-NAT within the stated experimental uncertainty:
We compare $\Delta H^0_{ev}(HCl)$ of 78.4 ± 11.4 and 69.6 ± 5.8 kJ/mol for α- and β-NAT (equations
(23) and (25)). On the other hand we have equality, perhaps fortuitously, between $E_{ev}(HCl)$
and $\Delta H^0_{ev}(HCl)$ for α-NAT following equations (22) and (23) which leads to the expected
conclusion that HCl accommodation on α-NAT is not an activated process which essentially
has zero activation energy similar to the situation for $HNO_3$ interacting with β-NAT. On the
other hand, this type of argument would lead to a negative activation energy for HCl
accommodation on β-NAT because the enthalpy of evaporation of HCl from β-NAT is
smaller than $E_{ev}(HCl)$ from β-NAT. However, the kinetic data of $J_{ev}(HCl)$ for β-NAT may be
affected by saturation of HCl uptake because experiments have been performed using the PV
admission. This situation may be similar to the kinetic results of $J_{ev}(H_2O)$ for β-NAT
displayed in Figure 10 that shows a significantly smaller value for $E_{ev}(H_2O)$ in PV vs. TO
experiments (52.1 vs. 75.5 kJ/mol, see also Table 3) whereas the saturation effect seems not
to affect the kinetic data for α-NAT. The anomalously large experimental uncertainty for
$HNO_3$ TO experiments on α-NAT displayed in Table 3 certainly has to do with the restricted
temperature interval over which we were able to monitor α-NAT before it converted to β-
NAT. This may be seen in the synoptic overview of the van't Hoff plots for $HNO_3$ interacting
with NAT displayed in Figure S4 of Supplement D. This restricted T range is also visible in
Figure 11A for $J_{ev}(HNO_3)$ from α-NAT.





## 5  Conclusions and Atmospheric Implications


In this study we have confirmed that exposure of ice films to $HNO_3$ vapor pressures at
temperatures akin to the ones found in the stratosphere leads to formation of NAT hydrates.
Of the two known forms of NAT, namely $\alpha$-NAT and $\beta$-NAT, the latter is the
thermodynamically stable one whereas metastable $\alpha$-NAT is likely to be of lesser importance
in the heterogeneous processes at UT/LS atmospherically relevant conditions.
$R_{ev}(H_2O)$ on $\alpha$-NAT and $\beta$-NAT films are different compared to the case of HCl/ice where
the evaporation of $H_2O$ is not influenced by the presence of adsorbed HCl on the ice and takes
place at a rate characteristic of pure ice. This has important implications on the lifetime of
atmospheric ice particles. Ice particles with adsorbed $HNO_3$ forming NAT have longer
lifetimes compared to ice particles with adsorbed HCl, being amorphous or crystalline
$HCl \cdot 6H_2O$. In light of our results we raise the question if HCl-containing ice particles are of
significant atmospheric relevance as substrates for heterogeneous reactions due to their
reduced lifetimes and concurrent reduced opportunities to enable heterogeneous atmospheric
reactions.
$J_{ev}(H_2O)$ on $\alpha$-NAT presents a discontinuity at 181 K akin to that observed in pure ice by
Delval and Rossi (2004) and Pratte et al. (2006). The resulting Arrhenius representation at
high temperatures larger than $181 \pm 2$ K:

$$\log J_{ev}(H_2O)[molec \cdot cm^{-2} \cdot s^{-1}] = (35.9 \pm 2.8) - \frac{(75.3 \pm 9.9) \times 10^3}{2.303\, RT}$$

$J_{ev}(H_2O)$ on $\beta$-NAT shows two values depending on the measurement techniques as a result of
the propensity of the PV experiment to saturate the gas-condensate interface. TO experiments
are less precise but more accurate owing to the fact that they are less prone to saturation
compared to PV experiments. Therefore, we report results of TO experiments as preferred
values, whereas we rule out kinetic PV results owing to possible saturation problems and note
in passing that $\beta$-NAT is apparently prone to saturation than $\alpha$-NAT. The Arrhenius
representation for the preferred TO results is:
TO Experiments:  $\log J_{ev}(H_2O)[molec \cdot cm^{-2} \cdot s^{-1}] = (36.0 \pm 1.3) - \frac{(77.0 \pm 4.9) \times 10^3}{2.303\, RT}$
HCl kinetic measurements on $\alpha$-NAT and $\beta$-NAT indicate that HCl does not displace a
significant number of $HNO_3$ molecules from the ice surface upon deposition, but rather that



HCl and HNO$_3$ do not strongly interact with each other in the condensed phase and that HCl
evaporates faster. This observation is also supported by the slower rates of evaporation and
the correspondingly higher values of the HNO$_3$ evaporation activation energy on α-NAT and
β-NAT, E$_{ev}$(HNO$_3$) = (178.0 ± 27.4) kJ mol$^{-1}$ and E$_{ev}$(HNO$_3$) = (102.0 ± 8.6) kJ mol$^{-1}$ (see
Table 3), respectively, compared to the activation energy for HCl evaporation on HCl•6H$_2$O,
E$_{ev}$(HCl) = (87.0 ± 17) kJ mol$^{-1}$. This also is consistent with a larger calculated interaction
energy of HNO$_3$ with H$_2$O ("affinity") in α-NAT compared to β-NAT (Weiss et al., 2016)
despite the fact that ΔH$_f^0$(α-NAT) is higher by 6 ± 20 kJ/mol compared to β-NAT.
The reliable and reproducible measurement of the vapor pressure of H$_2$O (P$_{H2O}$) in the UT/LS
still represents a thorny problem on airborne (aircraft and balloon) platforms owing to small
absolute values as well as to possible rapid variations as a function of altitude. Fahey and
coworkers have found an elegant way to solve this problem using a suitably adapted chilled
mirror hygrometer (CMH) where a cryogenic ice deposit on a temperature controlled mirror is
monitored during atmospheric sampling using a backreflected IR LED element that controls a
mirror heater in a feed-back loop (Thornberry et al., 2011). When **P$_{H_2O}$** increases the mirror
reflectivity decreases owing to a concomitant, but presumed increase in scattering because of
the formation of a polycrystalline ice deposit on the mirror. In this case the mirror heating
power is increased in a feedback loop in order to restore the original reflectivity at the former
operating conditions.
For concentrations of 1-10 ppm H$_2$O and 0.1-4 ppb HNO$_3$ typically encountered in this region
of the atmosphere (UT/LS) we expect a weak perturbation of the cryogenic ice deposit
through co-deposition of HNO$_3$ on the mirror. In fact, Thornberry et al. (2011) measure a
HNO$_3$ deposit from their laboratory experiment corresponding to roughly a molecular
monolayer on the 0.37 cm$^2$ mirror (geometric) surface at typically 4 ppb P$_{HNO3}$ and a total
deposition time of 3 h. This is a negligible quantity of HNO$_3$ compared to the 2000 or so ice
molecular bilayers per μm of ice deposited. Fahey and coworkers have recently introduced an
advanced version of a hygrometer that monitors gas phase H$_2$O using a high resolution diode
laser near 2694 nm at a specific H$_2$O absorption line (Thornberry et al., 2015). This
methodology replaces the unspecific monitoring of the broad-band reflectivity by an
identifiable spectroscopic molecular IR transition of gas phase H$_2$O and thus removes the
doubt about the identity of the absorber compared to the prior use of the (broad band) IR




LED. It is our understanding that this advanced CMH is in the process of actually being tested
in the field.
However, when the CMH was used in a laboratory flow reactor experiment at a higher
concentration of $H_2O$ and $HNO_3$ (both at typically 80 ppb) the mirror reflectivity increased
and led to an approximately 3 K lower mirror temperature at 194 K after approximately 4
hours into the experiment compared to a reference CMH not exposed to $HNO_3$. At first, the
authors identified the first $HNO_3/H_2O$ condensate as an $\alpha$-NAT coating on a $H_2O$ thin film
after approximately 1.8 hours into the co-deposition experiment of admitting six ppm $H_2O$
and 80 ppb $HNO_3$ into the flow reactor. At 2.3 hours after start the $HNO_3$ flow was halted
while maintaining a $H_2O$ flow of 80 ppm which led to the appearance of a "second
condensate" after approximately 4 hours of elapsed time. The authors attributed this "second
condensate" to an unknown $HNO_3/H_2O$ phase that led to a 63% supersaturation with respect
to pure ice corresponding to the above-mentioned 3 K depression of the mirror temperature.
It is perhaps useful to remind the reader at this point that the CMH detector of $P_{H_2O}$
compensates the change in reflectivity detected as a signal on a photodiode with a change in
mirror temperature, but the true molecular identity of the condensate goes unnoticed. Based
on the present results we claim that the selective evaporation of the heavier components
$HNO_3$ compared to $H_2O$ evaporation in the binary, and HCl in the ternary condensed phase
system is not possible, at least at atmospherically relevant $HNO_3$ and HCl concentrations
because $J_{ev}(HNO_3)$ and $J_{ev}(HCl)$ are always smaller than $J_{ev}(H_2O)$ for the investigated nitric
acid hydrates in the range 170-205 K. This means that the "second condensate" must still
contain some $HNO_3$ throughout the duration of the experiment. As far as positive proof for
the existence of an as yet unidentified $HNO_3/H_2O$ hydrate ("second condensate") is concerned
that results from the CMH-equipped flow experiment discussed above, we would like to
withhold judgement until possible consequences of the optical properties of the $HNO_3/H_2O$
condensate on the reported $H_2O$ supersaturation have been considered, including its temporal
changes at the chosen experimental conditions.
Using the real part of the index of refraction n at approximately 200 K of 1.333, 1.513 and
1.460 for pure $H_2O$ ice, $\alpha$- and $\beta$-NAT, respectively [Berland et al., 1994; Toon et al., 1994],
we calculate a specular reflectivity R of 2.0, 4.2 and 3.5% for pure $H_2O$ ice, $\alpha$- and $\beta$-NAT
following the Fresnel expression $(R = (n_0-n_1)^2/(n_0+n_1)^2)$ with $n_0$ referred to pure $H_2O$ ice) for
normal incidence. This shows that a potentially significant change of the optical properties of



a HNO$_3$-containing ice film relative to pure ice may be expected at these high doses of HNO$_3$,
which will critically depend on many geometric and molecular parameters including the
structure and concentration gradients of the film itself.
The implementation of a detailed (geometrical) optical model of the cryogenic film
interacting with the IR-emission is clearly beyond the scope of the present work, but it seems
judicious to take these optical changes into account in the future for the quantitative
interpretation of the experimental results of Gao et al. (2016). In addition, it will be necessary
to gauge the importance of film volume absorption as the emitted IR radiation will pass twice
across the film thickness on its way to the detector. We have recently measured the optical
cross sections of the nitric acid hydrates dealt with in the present work in the range 4000-750
cm$^{-1}$ (Iannarelli and Rossi, 2015) which completes the set of optical constants of the nitric
acid hydrates in the IR spectral region (Toon et al., 1994). In the end the decision on the
existence of an unknown/unidentified HNO$_3$/H$_2$O phase present in the UT/LS that
significantly exceeds the saturation vapor pressure of pure ice will probably hinge on
experiments performed using the advanced version of the hygrometer that is based on the
absorption of high resolution radiation near 2.7 μm by gas phase water vapor mentioned
above. This gas-phase measurement system seems free of perturbations by other atmospheric
gases and therefore is likely to be suitable in order to resolve the question at hand (Thornberry
et al., 2015).


**Acknowledgements**
The authors would like to acknowledge the generous support of this work over the years by
the Swiss National Science Foundation (SNSF) in the framework of projects 200020_125204
and 200020_144431/1. We also sincerely thank Mr. Alwin Frei of PSI for graciously granting
the permission to perform the experiments in his laboratory. In addition, our warm thanks go
to Mr. René Richter and Günther Wehrle for hardware and IT support so sorely needed.



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



Table 1: Characteristic parameters of the present flow reactor.

| | | | |
|---|---|---|---|
| Reactor volume (upper chamber) | $V_R = 2036$ cm$^3$ | | |
| MS (lower) chamber | $V_{MS} = 1750$ cm$^3$ | | |
| Reactor internal surface | $S_W = 1885$ cm$^2$ | | |
| H$_2$O calibrated volume – inlet line | $V_{water} = 62$ cm$^3$ | | |
| HNO$_3$ calibrated volume – inlet line | $V_{acid} = 20$ cm$^3$ | | |
| Si support area (one side) | $A_{Si} = 0.99$ cm$^2$ | | |
| Surface to Volume ratio | $2\,{A_{Si}}/{V_R} = 0.9725\times10^{-4}$ cm$^{-1}$ | | |
| Reactor wall temperature | $T_w = 315$ K | | |
| Conversion of evaporation rate and flux | $R_{ev} \cdot V_R = 2 \cdot A_{Si} \cdot J_{ev}$ | | |
| | **HNO$_3$** | **H$_2$O** | **HCl** |
| Base Peak Signal MS [m/z] | 46 | 18 | 36 |
| MS Calibration Factor $C^X$ [molec$^{-1}$ s A] | $4.53\times10^{-25}$ | $6.65\times10^{-25}$ | $1.30\times10^{-25}$ |
| Escape rate constant $k_{esc}^S = C^S\sqrt{\dfrac{T}{M}}$ (small orifice) [s$^{-1}$] | 0.0913 | 0.1710 | 0.1213 |
| $k_{esc}^M = C^M\sqrt{\dfrac{T}{M}}$ (both orifices) [s$^{-1}$] | 0.4331 | 0.8102 | 0.5729 |
| Gas-surface collision frequency at 315 K, one side [s$^{-1}$] [a] $\omega = \dfrac{\overline{c}}{4V}\cdot A_{Si} = \sqrt{\dfrac{8\,R\,T}{\pi\,M}}\cdot\dfrac{A_{Si}}{4V}$ | 3.95 | 7.39 | 5.22 |

[a] M in kg; $A_{Si}$ in m$^2$; V in m$^3$; R = 8.314 J K$^{-1}$ mol$^{-1}$. "One side" corresponds to front or rear
side of Si-window. In order to calculate the accommodation coefficient α using equation (3)
we have used 2ω as the total collision frequency for both sides of the Si-window.




Table 2: Fit parameters of the Langmuir adsorption isotherms for $H_2O$, $HNO_3$ and HCl
interaction with the internal stainless steel (SS304) surfaces of the reactor.

| Adsorbed Gas (Additional Gas) [a] | $K_L$ $[\times 10^{-14}]$ [b] | $N_{TOT}$ $[\times 10^{17}]$ [c] | $N_{MAX}$ $[\times 10^{14}]$ [d] | $\alpha_w$ $[\times 10^{-6}]$ [e] |
|---|---|---|---|---|
| $H_2O$ | $3.18 \pm 0.38$ | $7.03 \pm 0.42$ | $3.73 \pm 0.22$ | $6.19 \pm 0.08$ |
| $H_2O$ (HCl, $F_{in} = 8\times10^{14}$) | $4.67 \pm 0.39$ | $8.38 \pm 0.29$ | $4.45 \pm 0.15$ | – |
| $HNO_3$ | $1.10 \pm 0.16$ | $93 \pm 11$ | $49 \pm 6$ | $2.92 \pm 0.10$ |
| $HNO_3$ ($H_2O$, $F_{in} = 2\text{-}3\times10^{15}$) | $1.61 \pm 0.40$ | $76 \pm 15$ | $40 \pm 8$ | – |
| $HNO_3$ (average values) | $1.28 \pm 0.17$ | $84 \pm 8$ | $45 \pm 4$ | – |
| HCl | $437 \pm 21$ | $5.06 \pm 0.06$ | $2.68 \pm 0.03$ | $16.9 \pm 0.3$ |
| HCl ($H_2O$, $F_{in} = 6\times10^{15}$) | $63.1 \pm 4.9$ | $4.85 \pm 0.07$ | $2.57 \pm 0.04$ | – |
| HCl ($H_2O$, $F_{in} = 3\times10^{15}$) | $64.6 \pm 6.3$ | $3.79 \pm 0.09$ | $2.01 \pm 0.04$ | – |

[a] $F_{in}$ is the flow rate of the additional gas in molec s$^{-1}$.
[b] $K_L$ is the Langmuir adsorption equilibrium constant in cm$^3$ molec$^{-1}$.
[c] $N_{TOT}$ is the total number of adsorbed molecules onto the internal surfaces, reported is the
saturation value for total internal surface (1885 cm$^2$) of SFR.
[d] $N_{MAX}$ is the adsorption site density in molec cm$^{-2}$.
[e] $\alpha_w$ is the reactor wall accommodation coefficient.





Table 3: Synopsis of thermodynamic ($P_{eq}$) and kinetic ($J_{ev}$) parameters of the Arrhenius and
van 't Hoff representation of data from Figure 2, Figure 4, Figure 6 and Figure 7.

| | | | $J_{ev}$ [a] | | $P_{eq}$ [b] | |
| --- | --- | --- | --- | --- | --- | --- |
| | | | $E_{ev}$ | A | $\Delta H^0_{ev}$ | $\Delta S/R$ |
| | | | | | | |
| Sample | Gas | Exp. | | | | |
| α-NAT | H$_2$O | TO | 75.3 ± 9.9[c]<br>3.5 ± 4.2[c] | 35.9 ± 2.8 | 70.3 ± 14.1 | 15.2 ± 4.0 |
| | | PV | 75.3 ± 9.9[c]<br>3.5 ± 4.2[c] | 15.1 ± 1.2 | 56.5 ± 5.1 | 11.8 ± 1.5 |
| | HNO$_3$ | TO | 178.0 ± 27.4 | 62.3 ± 7.8 | 128.6 ± 42.4 | 29.3 ± 12.0 |
| | HCl | PV | 78.3 ± 19.2 | 34.8 ± 5.3 | 78.4 ± 11.4 | 15.7 ± 3.2 |
| β-NAT | H$_2$O | TO | 77.0 ± 4.9 | 36.0 ± 1.3 | 76.7 ± 17.7 | 16.7 ± 4.9 |
| | | PV | 52.1 ± 2.4 | 28.7 ± 0.7 | 75.5 ± 11.1 | 16.7 ± 3.0 |
| | HNO$_3$ | TO | 102.0 ± 8.6 | 40.6 ± 2.4 | 96.5 ± 12.0 | 19.8 ± 3.3 |
| | HCl | PV | 56.7 ± 4.6 | 28.6 ± 1.3 | 69.6 ± 5.8 | 13.3 ± 1.6 |


[a] for gas X, R = 8.314 J K$^{-1}$ mol$^{-1}$:  $\log J_{ev}(X)[\text{molec} \cdot \text{cm}^{-2} \cdot \text{s}^{-1}] = A - \dfrac{E_{ev} \times 10^3}{2.303 \, RT}$
[b] for gas X, R = 8.314 J K$^{-1}$ mol$^{-1}$:  $\log P_{ev}(X)[\text{Torr}] = \dfrac{\Delta S}{R} - \dfrac{\Delta H^0_{ev} \times 10^3}{2.303 \, RT}$
[c] Fit consists of two straight lines intersecting at 181 ± 2 K with two values for $E_{ev}$.



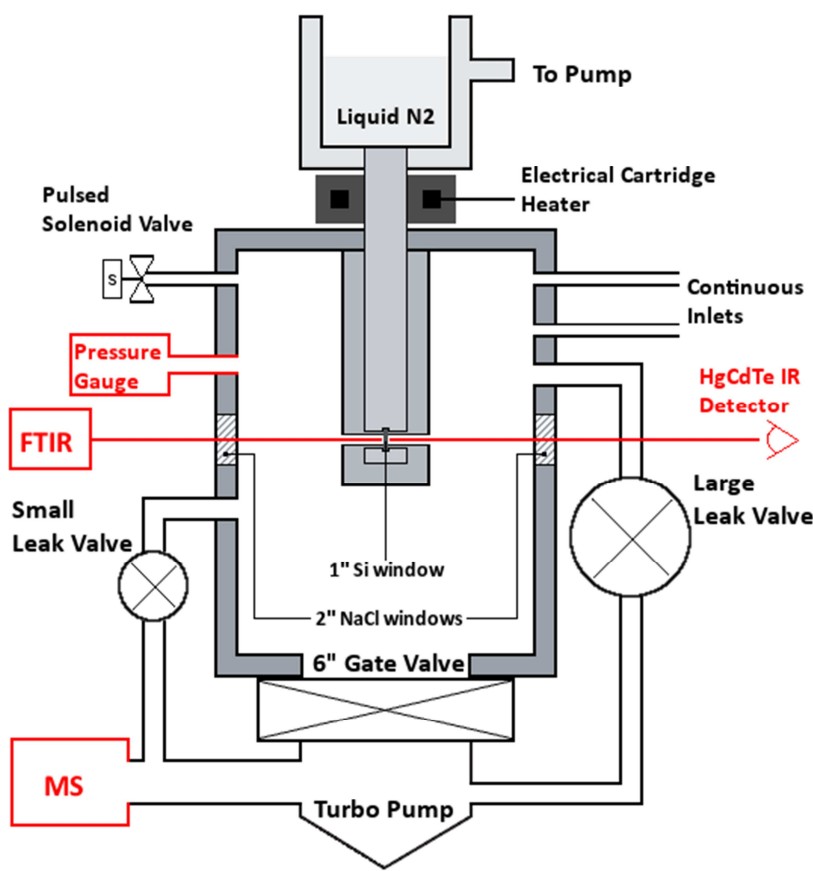


Figure 1: Schematic drawing of the reactor used in this work. The diagnostic tools are highlighted in red and important parameters are listed in Table 1 and Table 2. The ice film is deposited on both sides of the 1" diameter Si window (black vertical symbol hanging from cryostat inside reaction vessel).








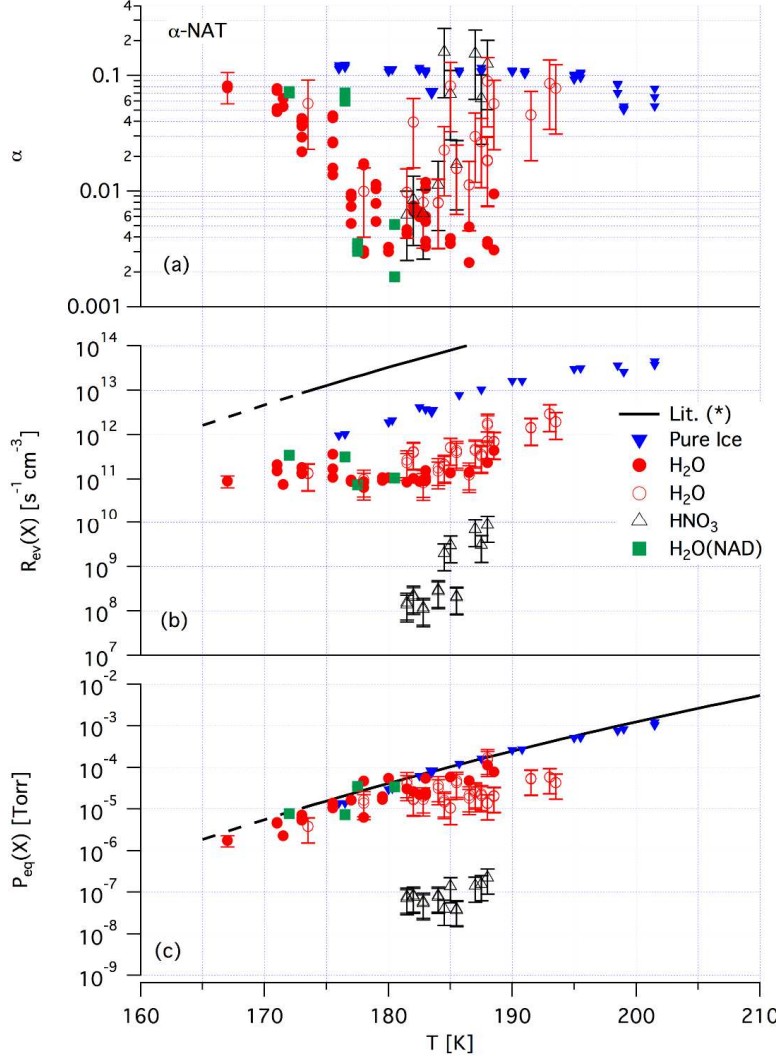


Figure 2: Synopsis of kinetic results for α-NAT and NAD using $H_2O$ as a probe gas in PV
experiments and $H_2O$ and $HNO_3$ in two-orifice (TO) experiments. Full symbols represent PV
experiments and empty symbols represent TO experiments. Further explanations of the used
symbols may be found in the text. The calculated relative error for PV experiments is 30%
whereas for TO experiments we estimate a relative error of 60%. Examples of the amplitude
of the errors are reported for selected points. The black line shows results from Marti and
Mauersberger (1993) with $R_{ev}(H_2O)$ of pure ice calculated for the system in use using α = 1.

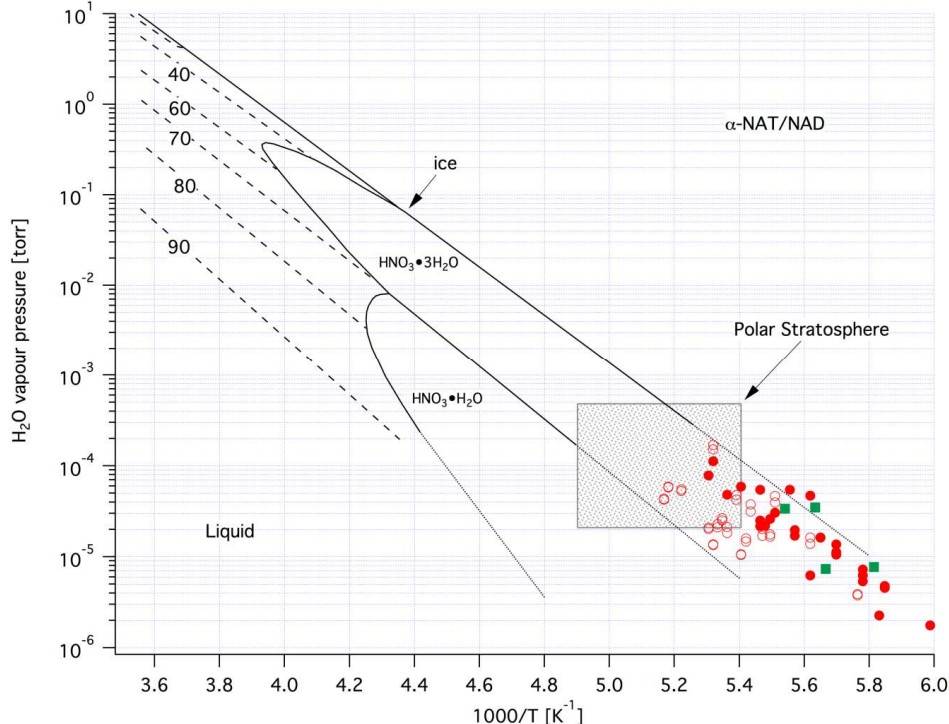


Figure 3: Binary phase diagram of the $HNO_3/H_2O$ system reconstructed from McElroy et al.

(1986); Hamill et al. (1988); Molina (1994). The full symbols represent calculated values of

$P_{eq}(H_2O)$ for $\alpha$-NAT and NAD using the kinetic data of PV experiments. Empty circles

represent calculated values of $P_{eq}(H_2O)$ for $\alpha$-NAT using the kinetic data of two-orifice

experiments. The solid lines represent the coexistence conditions for two phases and the

dashed lines represent vapor pressures of liquids with composition given as % (w/w) of

$HNO_3$. The shaded gray represents polar stratospheric conditions.





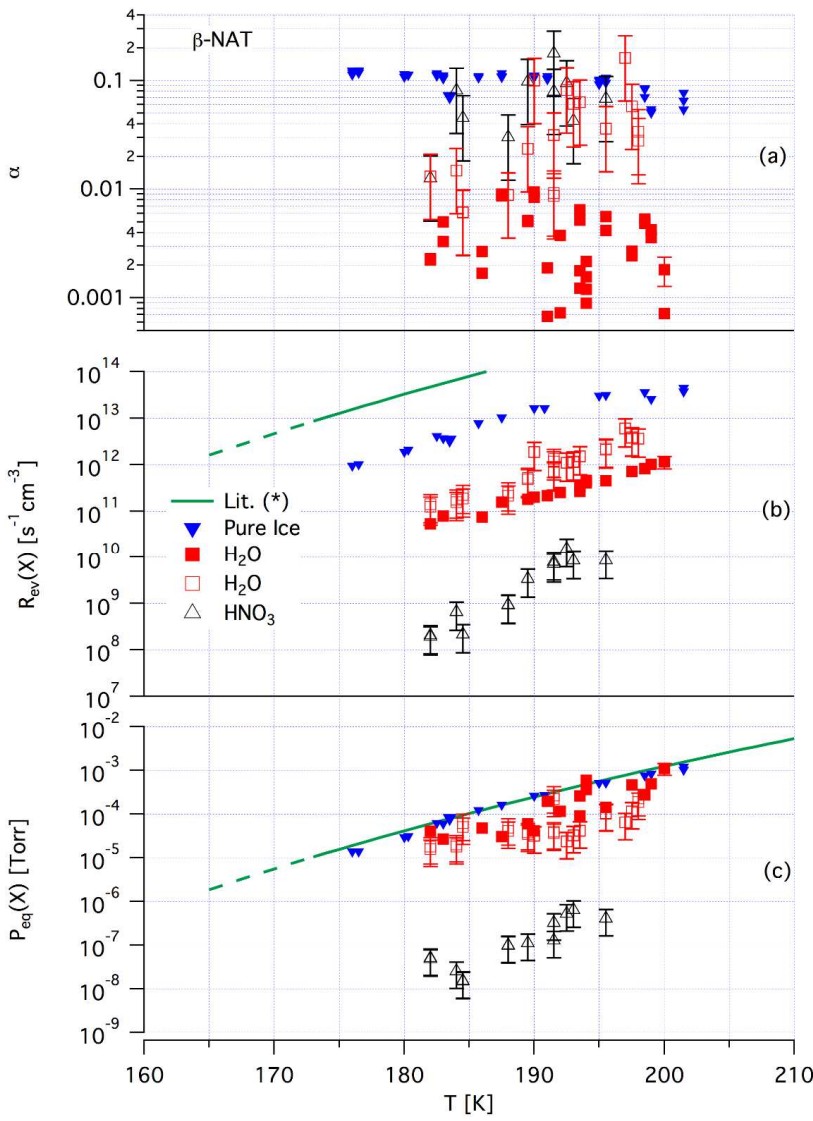


Figure 4: Synopsis of kinetic results for β-NAT using $H_2O$ as a probe gas in PV experiments
and $H_2O$ and $HNO_3$ in two-orifice experiments. Full symbols represent PV experiments and
empty symbols represent TO experiments. Further explanation of the used symbols may be
found in the text. The calculated relative error for PV experiments is 30% whereas for TO
experiments we estimate a relative error of 60%. Examples of the amplitude of the errors are
reported for selected points. The green line shows results from Marti and Mauersberger

1155  (1993).





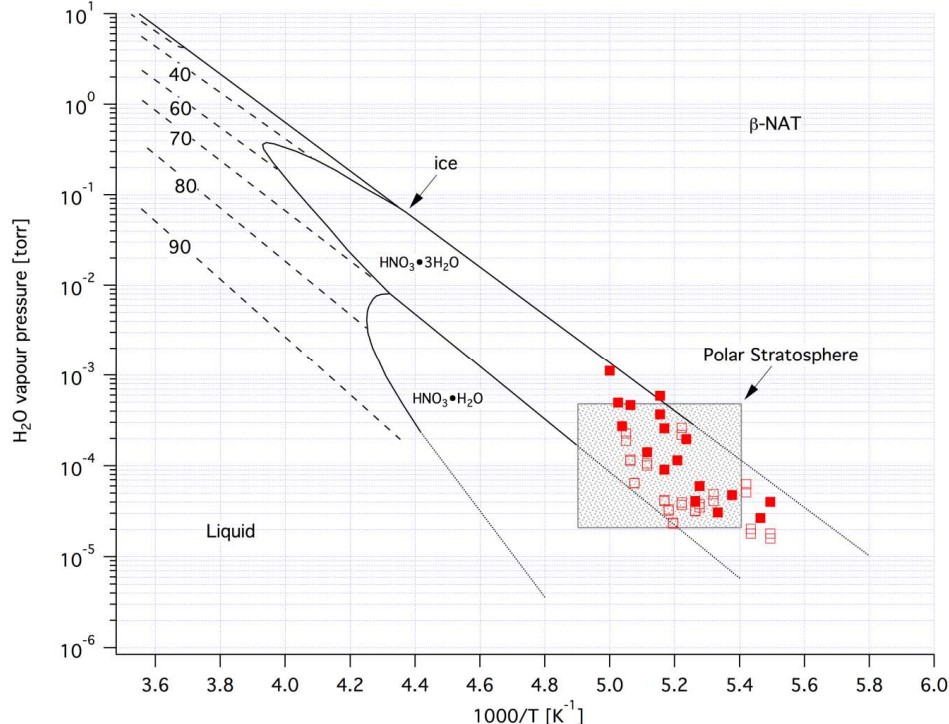


Figure 5: Binary phase diagram of the $HNO_3$/$H_2O$ system reconstructed from McElroy et al.

(1986); Hamill et al. (1988); Molina (1994). The full symbols represent calculated values of

$P_{eq}(H_2O)$ for β-NAT using the kinetic data of PV experiments. Empty circles represent

calculated values of $P_{eq}(H_2O)$ using the kinetic data of two-orifice experiments. The solid

lines represent the coexistence conditions for two phases and the dashed lines represent vapor

pressures of liquids with composition given as % (w/w) of $HNO_3$. The shaded gray represents

polar stratospheric conditions.



Figure 6: Synopsis of kinetic results for α-NAT using HCl as a probe gas in PV experiments.
The used symbols are explained in the text. The calculated relative error for PV experiments
is 30%. The black line shows results from Marti and Mauersberger (1993).








Figure 7: Synopsis of kinetic results for β-NAT using HCl as a probe gas in PV experiments.

The used symbols are explained in the text. The calculated relative error for PV experiments
is 30%. The black line shows results from Marti and Mauersberger (1993).




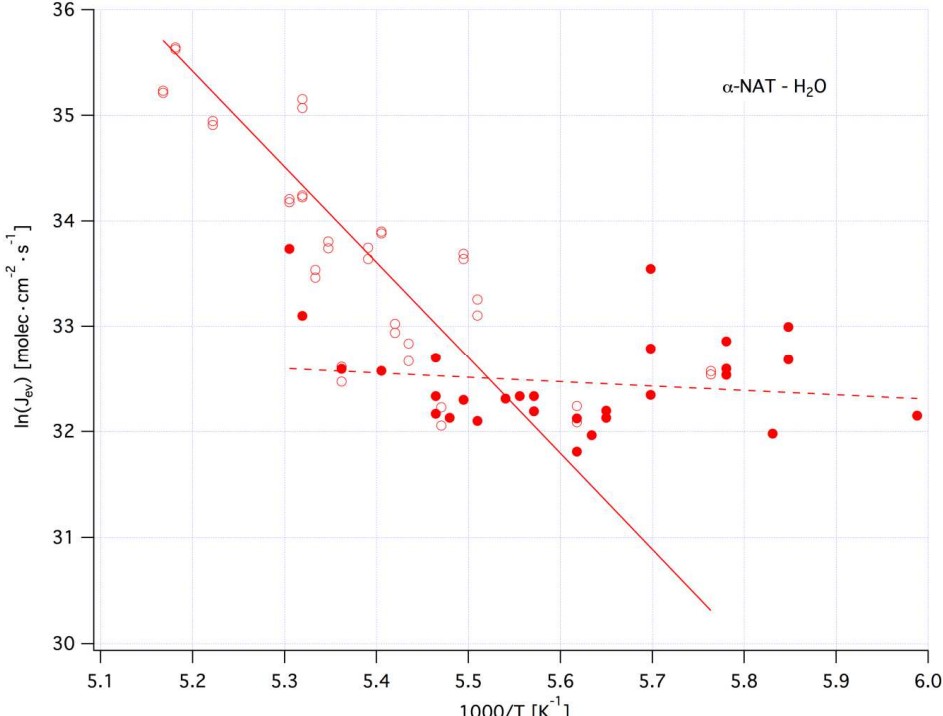


Figure 8: Arrhenius plot of $J_{ev}(H_2O)$ for α-NAT. Full and empty red circles represent results
of PV and TO experiments, respectively. Data are taken from Figure 2b and the equations for
the linear fits may be found in the text.





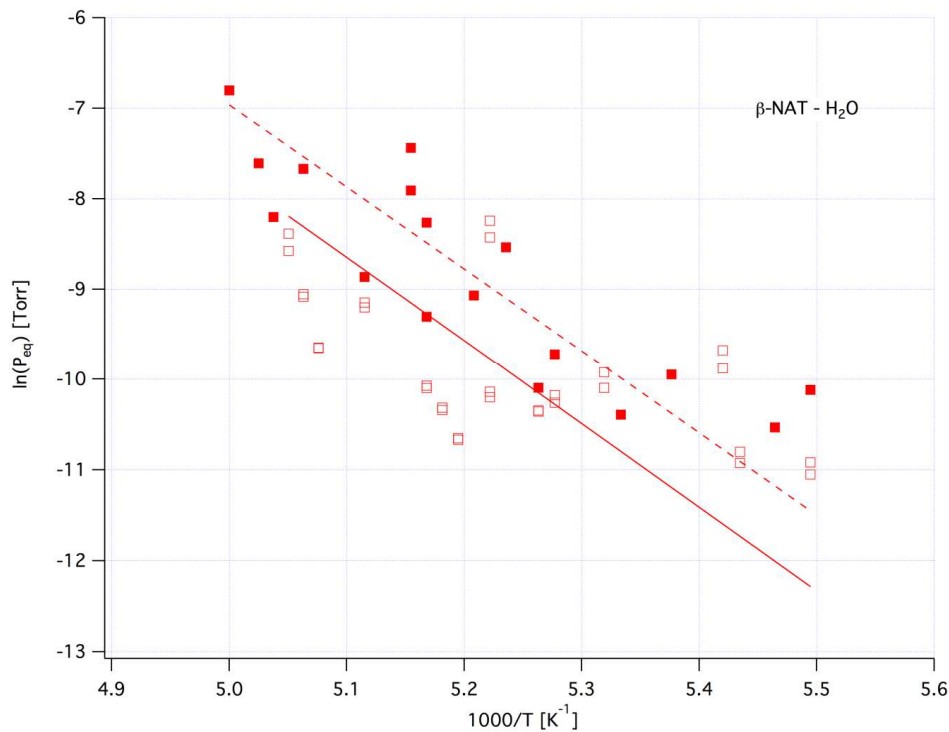


Figure 9: van 't Hoff plot of $P_{ev}(H_2O)$ for β-NAT data displayed in Figure 4c. Full and empty
red squares represent results of PV and TO experiments, respectively. The equations for the
linear fits may be found in the text.





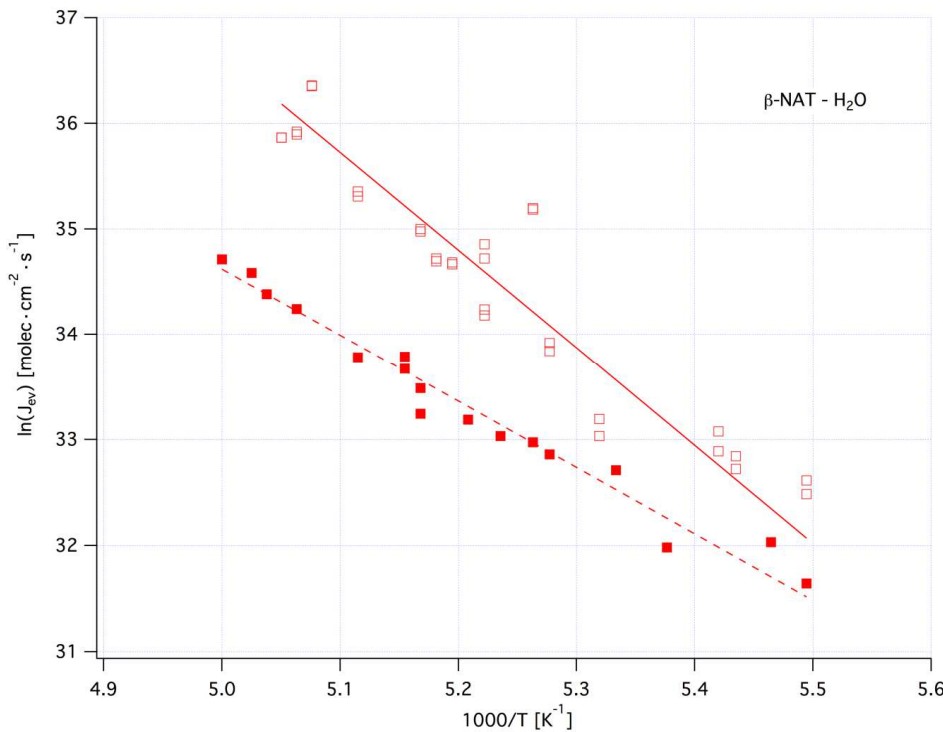


Figure 10: Arrhenius plot of $J_{ev}(H_2O)$ for β-NAT data displayed in Figure 4b. Full and empty
red squares represent results of PV and TO experiments, respectively. The equations for the
linear fits may be found in the text.





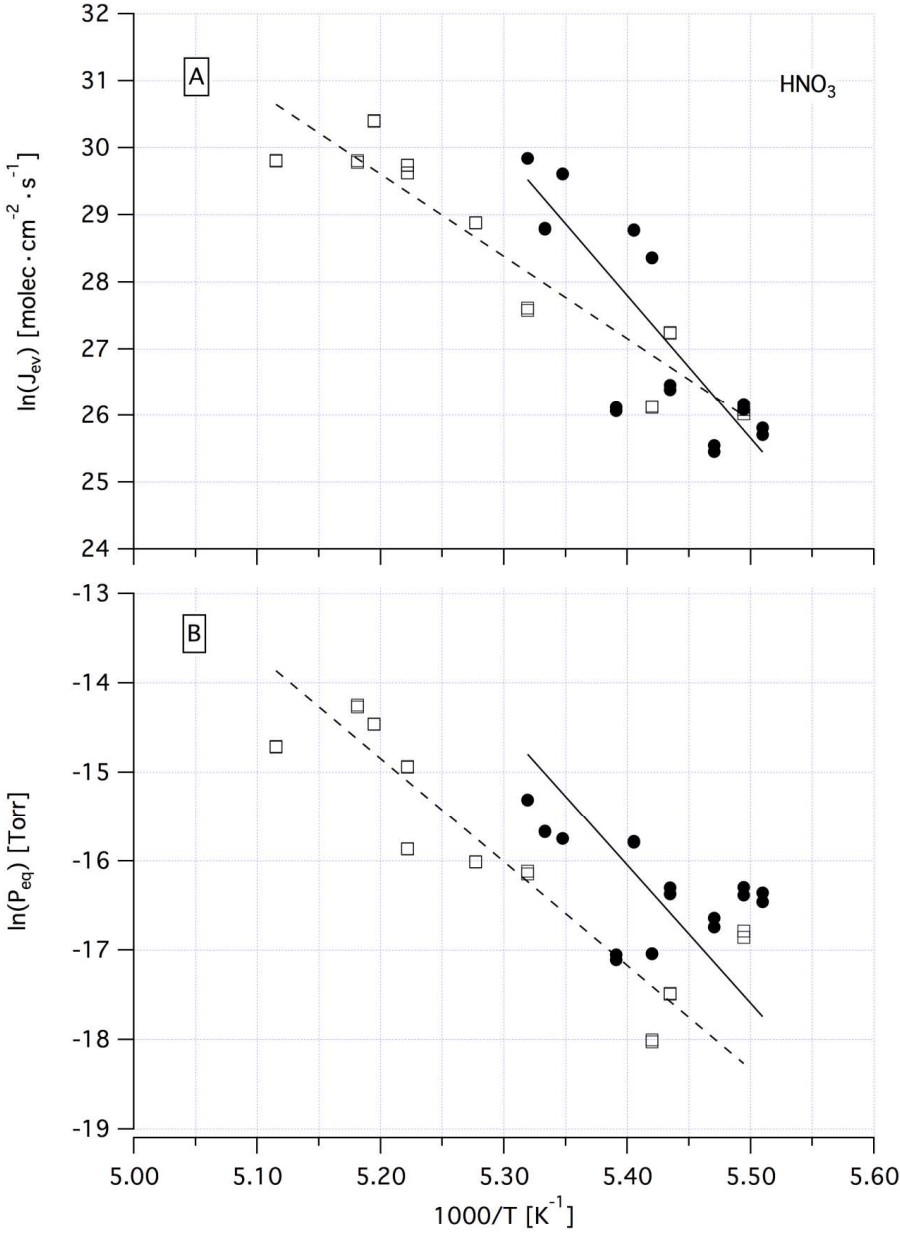


Figure 11: Arrhenius plot of $J_{ev}(HNO_3)$ (A) and van 't Hoff plot of $P_{eq}(HNO_3)$ (B) for α-NAT

(Figure 2b and Figure 2c) and β-NAT (Figure 4b and Figure 4c) resulting from TO

experiments. Full black circles and empty black squares represent the interaction of $HNO_3$

with α- and β-NAT films, respectively. The equations for the fitting lines may be found in the

text.



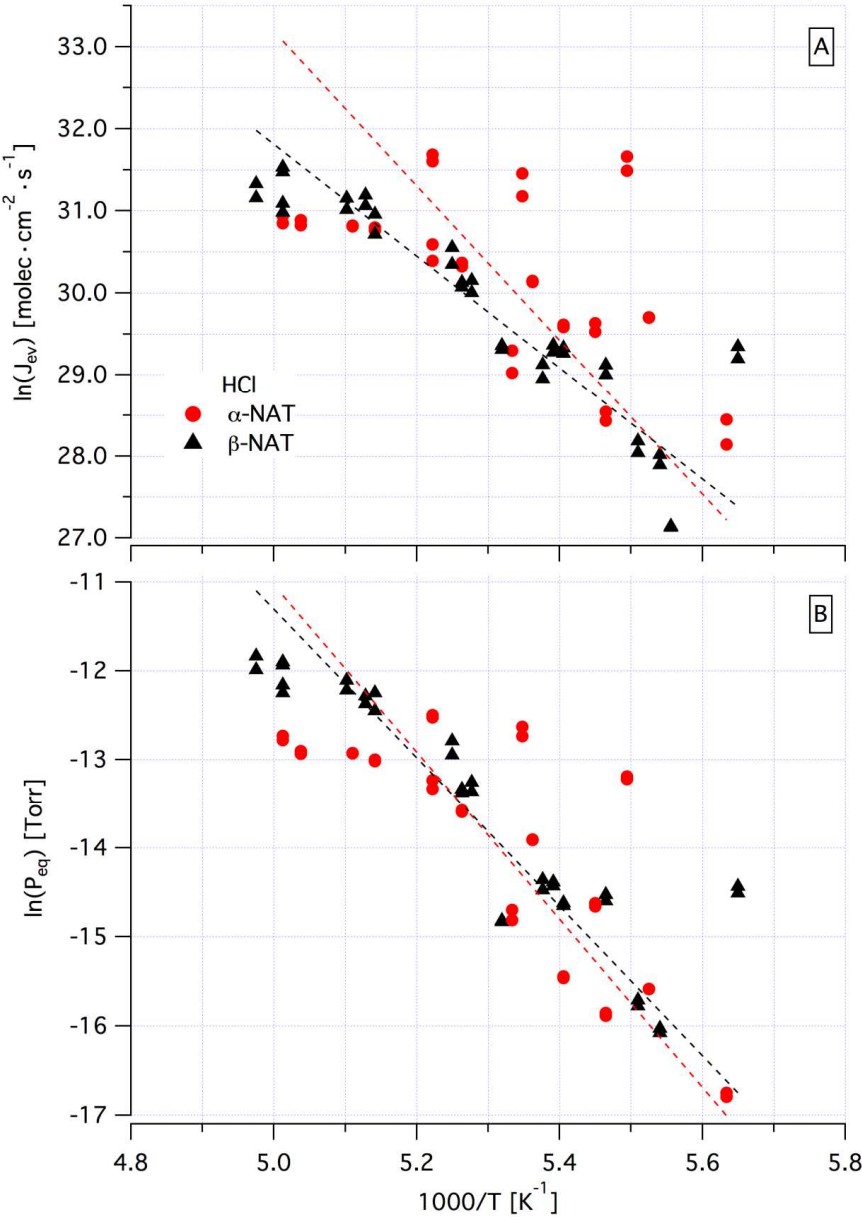

1190

Figure 12: Arrhenius plot of $J_{ev}$(HCl) (A) and van 't Hoff plot of $P_{eq}$(HCl) (B) for α-NAT (Figure 6b and Figure 6c) and β-NAT (Figure 7b and Figure 7c) resulting from PV experiments. Full red circles and black triangles represent the interaction of HCl with α- and β-NAT films, respectively. The equations for the fitting lines may be found in the text.