# Peer review of "Heterogeneous Kinetics of H2O, HNO3 and HCl on HNO3"

_Atmospheric Chemistry and Physics, 2016_

## Referee Comment (RC1) · Anonymous Referee #1 · 16 May 2016

Comments on the manuscript entitled: "Heterogeneous kinetics of H2O, HNO3 and HCl on HNO3 hydrates (ïĄą-NAT, ïĄć-NAT, NAD) in the range 175-200 K" Author(s): R. Iannarelli and M. J. Rossi MS No.: acp-2016-247

This paper reports the results of heterogeneous kinetics of H2O, HNO3 and HCl on HNO3 hydrates under stratospheric temperature conditions in the range of 175-200 K. Langmuir adsorption isotherms were used to take into account wall interactions for the aforementioned species. Experiments were performed using a combination of transmission FTIR spectroscopy and mass spectrometry where partial and total pressure measurements have been employed in order to monitor growth and evaporation processes as a function of temperature using both pulsed admission and continuous

monitoring using multi-diagnostic stirred flow reactor. The manuscript is well written and contains thorough sets of experiments that can add to the understanding of adsorption of gases on nitric acid hydrates under stratospheric conditions. I recommend the manuscript be published following the authors response to the following comments.
.

Specific comments: 1- Page 1, line 19, FTIR absorption spectroscopy and not spectrometry 2- Page 7, line 185, silicon has a cutoff of 1500 cm-1 in the FTIR so how is the range extends from 4000-700 cm-1 3- Page 8, lines 219-220, the authors discuss that the transition in phases was observed via FTIR yet no FTIR or MS spectra were shown in the entire 52 pages of the manuscript. It would be interesting to the readers to show sample spectra and also to mention in a table the m/z and the wavenumbers where hydrates, HNO3, HCl and water were observed. 4- Page 13, lines 369-372, the authors discussed the difference between ïĄą-NAT and HCl; yet no HCl results were shown in figure 2 5- Page 14, line 421, can the authors comment how the relative errors were calculated and why same error in PV (30%) and TO (60%) experiments were observed on both the NAT and NAD films? 6- Page 15, lines 448-453, again the authors talk about comparisons to HCl experiments however no HCl data are present in Figure 4b. Which figure the authors want the reader to check to compare HCl case to figure 4a, please mention the figure since HCl experiments are introduced in the next section 7- Page 17, lines 484-488, why are the authors making assumptions regarding the substrates can't they get information on changes due to HCl from FTIR? 8- Page 18, lines 534-535, the authors mentioned a decrease in ïĄą-ïĄć-NAT as a function of increasing temperature but looking at figure 7a it looks like there was no change in the signal within experimental error 9- Page 19, lines 563-574 are the two distinct temperature regimes in Figure 2a due to surface disorder on ice? 10- Page 24, lines 704-709 why only TO experiments were possible for HNO3? This point is not so clear 11- Page 25, lines 753-758 can the authors comment why their results for HCl experiments were different from those by Haynes (2002)? 12- Figures 2-7 although the

authors mentioned the symbols in the text but it was so confusing to keep going back and forth between the text and the figure given the extra length of this manuscript and the different systems studied. I recommend that the authors explain the symbols in the caption for every figure.

---

## Short Comment (SC1) · 31 May 2016

Dear Colleagues,

We read with interest the results of your recent experiments investigating the kinetics of H$_2$O and HNO$_3$ (and HCl) on NAT. We would like to offer some comments on the portion of the manuscript that discusses chilled mirror hygrometers and our recent manuscript "Persistent Water-Nitric Acid Condensate with Saturation Vapor Pressure Greater than That of Hexagonal Ice"

Lines 855-865: Initial development of a frost-point hygrometer for measurement of water vapor in the UTLS was reported by Brewer et al. (1948). Numerous papers in

the scientific literature from the 1950s to present describe further instrumental developments and variations, characterize performance, and analyze results from chilled mirror hygrometer measurements of UTLS $H_2O$. Chilled mirror hygrometers have been used for decades on balloon sondes at locations around the world (e.g. Hurst et al., 2011). The instruments used in our studies are laboratory versions of the CFH (cryogenic frostpoint hygrometer) described by Vömel et al. (2007). Thornberry et al. (2011) describes details of the implementation of the CFH for our specific experiments.

Line 862-863: "increase in scattering because of the formation of a polycrystalline ice deposit" would be better described as "increase in scattering due to growth of the polycrystalline ice deposit"

Lines 866-872: The relevant conclusion from Thornberry et al. (2011) is that the lack of perturbation to the CMH temperature (relative to the control) from the accumulation of $HNO_3$ on the mirror implies that there was no change in the $H_2O$ saturation vapor pressure over the mirror condensate. This is reasonable, and not inconsistent with the results presented in the manuscript, given that the small amount of condensed $HNO_3$ and the high surface area of the polycrystalline frost result in incomplete frost surface coverage by $HNO_3$.

Lines 872-879: The TDL-based hygrometer that we recently developed for UTLS measurements (Thornberry et al., 2015) is not related to our laboratory frostpoint studies. As noted, it uses absorption spectroscopy to measure gas-phase $H_2O$ (as do a number of other hygrometers used on research aircraft, e.g. JLH, DLH, VCSEL, HAI) and is not relevant to the present discussion and we suggest that this paragraph be omitted to avoid confusion. If you continue to include this discussion, we suggest 'Fahey and coworkers' be replaced by 'Thornberry, Rollins and coworkers'.

Lines 880-890: To clarify the typical experimental sequence in our $HNO_3$-ice experiments using the frostpoint hygrometer reported in Gao et al., (2016):

1. Pure ice frost layers were established on both CMH mirrors in a low-pressure flow

with a stable $H_2O$ mixing ratio < 10 ppm.

2. $HNO_3$ was added to the flow between the two CMHs such that the flow past the second mirror contained 80-100 ppb $HNO_3$.

3. After a period of time (~1 hour in the experiment shown in Gao et al.), the temperature of the exposed mirror began to increase. The deviation increased with continued $HNO_3$ exposure until it stabilized at a temperature similar to the saturation temperature of NAT for the $H_2O$ and $HNO_3$ partial pressures in the flow.

4. When the mirror temperature was stable, the $H_2O$ mixing ratio in the flow (past both mirrors) was increased (to ~80 ppm in the experiment shown in Gao et al.) causing the temperatures of both mirrors to increase. The temperature of the $HNO_3$-exposed mirror stabilized at the NAT saturation temperature for the new $H_2O$ partial pressure.

5. The $HNO_3$ addition was shut off. $HNO_3$ in the flow dropped by an order of magnitude in the first two minutes and then continued to decrease more slowly (double exponential decay with time constants of several minutes and greater than 30 minutes). The $HNO_3$ mixing ratio in the flow continued to decrease, but remained above the background level for the length of the experiments. The temperature of the $HNO_3$-exposed mirror began to decrease immediately upon removal of the $HNO_3$ addition, but more slowly than the calculated NAT saturation temperature, indicating a kinetic limit to the response of the condensate to $HNO_3$ in the gas phase. After ~1 hour, the temperature of the exposed mirror reached that of the control (unexposed) mirror, ~213 K in the experiment shown in Gao et al.

6. Instead of restabilizing when it reached the pure water ice saturation temperature, the temperature of the exposed mirror continued to decrease, suggesting that the $H_2O$ vapor pressure of the condensate had become larger than that of pure $H_2O$ ice. The depression in the mirror temperature relative to the control was sustained though multiple changes in $H_2O$ amounts in the flow that caused the temperatures of both mirrors to increase or decrease by more than 5 °C, indicating that the condensate properties

were stable even following loss and uptake of water. The $HNO_3$ measured downstream of the mirror varied with the mirror temperature (the temperature of other surfaces in the system remained unchanged), providing additional evidence of the continued evolution of $HNO_3$ from the mirror. Forced evaporation of the condensate in some experiments confirmed the continued presence of $HNO_3$ on the mirror.

Lines 891-899: In Gao et al. (2016), we make no statements about the relative evaporation rates of $H_2O$ and $HNO_3$ from the mirror condensate (initially NAT), but argue that there is a continuing loss of $HNO_3$ from the condensate relative to $H_2O$ due to the lack of $HNO_3$ in the air flow while the $H_2O$ flux from the mirror is balanced by a flux to the condensate from the gas phase. We argue that the elevated vapor pressure of the condensate is indeed due to the presence of residual $HNO_3$ in the condensate that prevents the complete recrystallization of mirror $H_2O$ into hexagonal ice.

Lines 899-920: It is likely that changes in the refractive index due to changes in condensate composition as well as changes in condensate morphology (crystal size and shape and surface roughness) play a role in changing the relationship between the mass of condensate on the mirror and the diffuse reflectivity. However, the frostpoint hygrometer is inherently a steady state instrument, and when a stable relationship between the mass of the condensate and the reflectivity exists, be that pure hexagonal ice, NAT, or another form, the mirror reflectivity-temperature feedback will operate to maintain a constant mass on the mirror and the mirror temperature will therefore reflect the saturation vapor pressure of the reflection-controlling condensate. This can be seen in the transition of the ice frost layer to NAT in the first part of our laboratory experiments where the mirror temperature rises from ice saturation to that of NAT, but then is stable at the NAT saturation temperature regardless of the difference in refractive index between ice and NAT. For the $HNO_3$-$H_2O$ condensate with a lower $H_2O$ saturation temperature than pure ice that occurs in our experiments following the loss of some fraction of the $HNO_3$ from the NAT frost, the mirror temperature transitions to the lower value over a period of hours and then remains stable for many more hours, even as

$HNO_3$ continues to evolve from the mirror condensate. Additionally, the experiments in which the water vapor in the flow over the condensate was changed demonstrate the $H_2O$ vapor pressure control of the condensate as well as the stability of the condensate over a range of temperatures. Changes in condensate optical properties due to changes in composition or morphology that produce a change in the reflectivity would be interpreted by the CMH feedback control as a change in condensate mass and drive a compensating change in mirror temperature to restore the reflectivity to its set value. The temperature perturbation would only last until the resulting change in the condensate mass restored the original reflectivity, after which the mirror temperature would adjust to the saturation temperature necessary to maintain the condensate in steady state. Only a continuous change in the optical properties of the mirror condensate could produce an extended offset in the mirror temperature from the condensate saturation temperature.

Regards,

Troy Thornberry, Andrew Rollins, Ru-Shan Gao and David Fahey

*Additional References*

Brewer, A. W. and B. Cwilong and G. M. B. Dobson (1948), Measurement of Absolute Humidity in Extremely Dry Air, Proc. Phys. Soc., 60, 52-70.

Hurst, D. F., S. J. Oltmans, H. Vömel, K. H. Rosenlof, S. M. Davis, E. A. Ray, E. G. Hall and A. F. Jordan (2011) Stratospheric water vapor trends over Boulder, Colorado: Analysis of the 30 year Boulder record, J. Geophys. Res.-Atmos., 116, D02306, doi:10.1029/2010JD015065.

Vömel, H., D. E. David, and K. Smith (2007), Accuracy of tropospheric and stratospheric water vapor measurements by the cryogenic frost point hygrometer: Instrumental details and observations, J. Geophys. Res.-Atmos., 112, D08306, doi:10.1029/2006JD007224.

---

## Referee Comment (RC2) · Anonymous Referee #2 · 8 Jun 2016

Review of Iannarelli and Michel J. Rossi, ACPD

Overall: This laboratory study examines the kinetics of water vapor, nitric acid, and hydrochloric acid on nitric acid trihydrate films using two different approaches to examine surface chemistry. There are some discrepancies or incomplete datasets in the literature, and the strength of this study is the range and complementary techniques in which were used to study the kinetics. Overall, I find the experiments to be well conducted (with one exception, noted below). However, the organization and motivation needs to be made clearer, both in the introduction and in the atmospheric implications. Both sections read like a "data dump" with little explanation to identify the key discrepancies or limitations in the literature. Why are the authors conducting this study, 20+ years after some of the initial studies were conducted? Instead, the intro leads with a nice (but unnecessary) review of general PSC chemistry, something that is now several decades old and the overview of which is not necessary. The atmospheric implications section goes on a tangent (incorrectly, at that) on water vapor measurement instruments that really aren't related to the current study results. Both of these aspects distract the reader from the high-quality, laboratory study and their results. While this paper will be eventually publishable, it requires some significant revisions in its current form.

Key points: Lines 56-129: a review of PSC chemistry has been common knowledge for decades; this section reads like a review article and is not necessary for the manuscript; indeed, it distracts from the critical questions that this study is trying to examine.

Lines 130-157: While this is a thorough review of the literature, it reads to some extent like a "data dump". The experiments were done under different conditions to some extent. Are there real discrepancies between these results? Perhaps a table of past literature and your results would be more clear/helpful. At the very least, one should summarize the point of this section: e.g. there are discrepancies, there may or may not, too hard to say given the different experimental conditions, etc. and whatever it is, this is the motivation for our study! As written, the reader is left to search through a lot of data with no clear idea on whether there is true disagreement or not. And then explicitly tell what aspect of the study will your work address in this regard.

Paragraph 158-162: Now suddenly the authors switch their literature review to HNO3 on pure ice. Only the last two sentences of this paragraph seem relevant to the work, at least for the introduction. And even then, there should be a transitional statement such as "The complications/discrepancies of HNO3 and H2O update on NAT surfaces is also evident when examining HCl uptake on NAT." or similar

Line 192+: The authors mention that the inlet system was modified but then failed to even provide a brief sentence or two on the actual modification. If it is important to mention at the start, please briefly summarize the modification.

Line 222-224: Not entirely an apples-to-apples comparison. The RAIR study was most sensitive to very thin films (< 10s-100 nm) and the very near surface properties. At thicker films and higher dose rates, they observed similar results as past studies and even the current study in the manuscript. The technique in the manuscript most likely was not sensitive to the presence of very thin films that were observed in the RAIR study.

Line 236-239: This is one of my main concerns experimentally about the study. The excess ice, even if it stabilizes NAT, will impact the vapor pressures of water inside the chamber. Is one always on the ice-NAT phase line? If not, to what extent does each phase determine the partial pressure of water observed in the chamber? This has implications for the later results for the accuracy of H2O partial/vapor pressures and H2O kinetics. Why not just make a pure NAT film like to past thin film studies? NAT is a pretty stable film to make. The excess ice phase present needs to be discussed in more detail and a logical, reasoned argument why it doesn't complicate the interpretation of the results (or to what extent it does).

Lines 624-633: Report the entropies of evaporation as well – do they make sense with physical principles? If not, why? And elsewhere in the manuscript.

Line 660-661: Could a similar explanation be used to invoke discrepancies between your results and those in the literature (JPL recommendations)?

Line 681: Can the absolute value of 12 kJ/mole be explained physically in terms of hydrogen bonds? Why or why not?

Line 686-7: Warshawsky et al. GRL 1999 also quantified this process of a sealing NAT layer slowing ice evaporation, and these were done a much lower HNO3 partial pressures than in the Biermann et al. study. Related to this, what are the partial pressures of HNO3 used in these experiments? Are they relevant to the atmosphere at all? They seem like they were much higher than what is expected in the atmosphere based upon the discussion and comparison to other laboratory studies. Please cite the

HNO3 partial pressures used in these experiments.

Atmospheric implications: There is a data dump of numbers here once again, many of which were already described in detail in the discussion section. What are the key points, circling back to the motivation in the introduction and past literature experiments? e.g. Does the JPL kinetic data need to be revised (as suggested in the discussion in several places)? What are the implications of these much different values? How has this study broadened the range of past studies or explained potential discrepancies or unanswered questions in the past literature? What future research is needed? etc.

Also, the discussion on NAT-coated ice impacting field measurements is speculative, unsupported, and shows several large gaps of awareness in UTLS water vapor measurements. First, the authors cite the problems of "reliable and reproducible measurements" of water vapor in the field UTLS measurements. However, as noted above, I have serious questions on how one can reliably interpret the accuracy of the water vapor measurements in their laboratory setup given that two phases exist at conditions well off the ice/NAT equilibrium line (and higher HNO3 partial pressures than usually exist in the UTLS) – so it isn't clear to me how these laboratory results are that representative of the UTLS itself. Second, the CU/NOAA chilled mirror hygrometer has a long measurement history and is best described most recently by the Vömel et al. JGR, 2007 and/or Vömel et al., AMTD, 2016. More importantly, it compared extremely well in recent intercomparison campaigns to the reference standard (see Fahey et al. AMT 2014), an instrument/technique that probably is (in this reviewers' opinion) the most accurate/uncertainty-documented H2O measurement in the community. Third, HNO3 has not been shown in the NOAA tests to impact the frost layer (ice vapor pressure) at relevant HNO3 concentrations (Thornberry et al., AMT 2011). Fourth, there are numerous diode laser-based hygrometers by many leading groups in the world; in fact, I would argue the NOAA TDL is one of the most recent and, though promising and a quality measurement, has some of the least amount of field data to

characterize its strengths and weaknesses. More recent AquaVIT2 UT/LS water vapor intercomparisons showed some improved agreement in general from most of the UTLS hygrometers, whether diode laser-based at any wavelength (1.3, 1.4, 2.6 microns), laser-induced fluorescence, chilled mirrors, or other techniques. Therefore, I'm not sure the authors' results are applicable to explaining whether or not an instrument may work with the limited knowledge of the measurement instruments themselves and better agreement now being observed. This is especially true since the manuscript's lab results appear to be at HNO3 concentrations/thicknesses well above what is possible in the UTLS. The Gao et al. 2016 JPC-A dealt with very small amounts of residual HNO3 within ice and not related to thick NAT coatings here. For all of these reasons, I suggest removing these paragraphs on H2O measurements and expanding on the kinetics and the implications thereof/discrepancies.

Syntax/grammar/minor points: Line 135: don't need "respectively"

Line 436: A possible reason (singular)

---

## Author Comment (AC1) · 27 Jul 2016

Riccardo Iannarelli and Michel J. Rossi

michel.rossi@psi.ch

Answers to Question of Referee 1:

Q2- Page 7, line 185, silicon has a cutoff of 1500 cm-1 in the FTIR so how can the range extend from 4000-700 cm-1?

See Figure 1a, 1b.

Fig. 1a: Absorption spectrum of Si window (commercially available material from Nicodom sro)

Fig. 1b: Taken from the Handbook of Optics (Optical Society of America, McGraw-Hill

book Co. 1978)

A2- Figures 1a and 1b present transmission curves of Si windows that always have a very thin coating (on the order of 50-100 nm) SiO2 that protects the bulk of Si from oxidation. Although transmission is reduced in the 1500 to 600 nm range it is sufficiently transmitting to enable high-quality absorption spectra to be recorded. In our case the DIGILAB FTS 575 provides high throughput thanks to its 3" collection optics. The centerburst signal reduces from 9V to 3V after passage across 2a pair of 5mm thick KCl and a 2.0 mm thick Si window with external location of the HgCdTe detector cooled at 77 K.

Q3- Page 8, lines 219-220, the authors discuss that the transition in phases was observed via FTIR yet no FTIR or MS spectra were shown in the entire 52 pages of the manuscript. It would be interesting to the readers to show sample spectra and also to mention in a table the m/z and the wavenumbers where hydrates, HNO3, HCl and water were observed.

A3- We agree with the referee regarding the presentation of raw FTIR/MS data of the discussed ternary HNO3/HCl/H2O chemical systems. To this effect we have added two new Figures (6 and 7) displaying combined FTIR/MS sample data as well as corresponding Table 3. However, for the binary system HNO3/H2O we have presented the corresponding combined sample FTIR absorption/MS data already in the Iannarelli and Rossi (2015) publication (J. Geophys. Res. 120, 11707-11727, 2015) such that renewed presentation in the present context would appear not to be appropriate. We therefore point out this reference when discussing the thermodynamic and kinetic data of the simple binary HNO3/H2O system.

TEXT- We refrain at this point from showing raw data (FTIR absorption spectra and MS data as a function of time) because representative samples have been shown by Iannarelli and Rossi (2015) for alpha- and beta-NAT. We will defer the presentation of raw data on the interaction of HCl on alpha- and beta-NAT to Section 3.3 below.

The following text is introduced into chapter 3.3 which introduces the ternary HCl/HNO3/H2O system.

[revised manuscript text omitted]

Q4- Page 13, lines 369-372, the authors discussed the difference between Alpha-NAT and HCl; yet no HCl results were shown in Figure 2.

A4- The purpose of that statement regarding the difference between Rev(H2O) in the HCl vs. the HNO3 hydrate was to alert the reader to a significant difference between the two hydrates. We have inserted the two references that deal with the HCl hydrates (amorphous HCl hydrate and HCl Hexahydrate).

TXT- This result is very different compared to the previously studied case of HCl amorphous and crystalline hexahydrate using the same apparatus (Iannarelli and Rossi, 2013), where the evaporation of H2O takes place at a rate characteristic of pure ice despite the presence of adsorbed HCl on the ice and is in agreement with the findings of Delval and Rossi (2005).

Q5- Page 14, line 421, can the authors comment how the relative errors were calculated and why same error in PV (30%) and TO (60%) experiments were observed on both the NAT and NAD films?

A5- Although preferred from the point of view of avoiding sample saturation, we attribute twice the uncertainty to the TO compared to the PV technique. TO involves taking a difference of two large numbers in the denominator of Equations (7) and (8), which is the reason to attribute a larger experimental uncertainty to this method.

TXT- The largest uncertainty in our experiment is that of the flow rate introduced into the reactor, which is assigned a relative error of 25%. The flow rate measurement affects the calibration of the MS and therefore the measurement of all the concentrations in

the reactor (Eq. 4). Therefore, we estimate a global relative error of 30% for PV experiments and double this uncertainty for TO experiments because Equations (7) and (8) imply a difference of two large numbers in many cases, as discussed above. We therefore assign a global 60% relative error to results obtained in TO experiments.

Q6- Page 15, lines 448-453, again the authors talk about comparisons to HCl experiments however no HCl data are present in Figure 4b. Which figure the authors want the reader to check to compare HCl case to figure 4a, please mention the figure since HCl experiments are introduced in the next Section.

A6- As discussed above for alpha-NAT we are referring to a previous study on the BINARY HCl/H2O phase (Iannarelli and Rossi, 2013) whereas chapter 3.3 below deals with the TERNARY HCl/HNO3/H2O system.

TXT- As in the case of alpha-NAT, this result is very different compared to the case of HCl hydrates studied before using the same apparatus (Iannarelli and Rossi, 2013) where the evaporation of H2O is not influenced by the presence of adsorbed HCl on the ice and takes place at a rate characteristic of pure ice for all HCl concentrations used.

Q7- Page 17, lines 484-488, why are the authors making assumptions regarding the substrates can't they get information on changes due to HCl from FTIR?

A7- In response to your discussion point 3 above we have introduced Figures 6 and 7 displaying FTIR absorption spectra in the presence of HCl whose principal peak positions have been collected in the new Table 3 (not reproduced here but included in the new manuscript version). Regarding the ternary HCl/HNO3/H2O system treated here we had to make some verified assumptions in order to keep the experimental parameter space to an acceptable level. All three simplifying assumptions have been verified in the current laboratory experiments.

TXT- In order to restrain the number of independent measurements on this ternary

system to a practical level we had to make some assumptions and/or simplifications in order to measure the unknown parameters of Eq. (2) for each gas used. Specifically, we made the following reasonable assumptions, both for alpha-NAT and beta-NAT substrates which have been experimentally verified in laboratory experiments:

Q8- Page 18, lines 534-535, the authors mentioned a decrease in ïAËŻaËŻ-ïAËŻc′-NAT as a function of increasing temperature but looking at figure 7a it looks like there was no change in the signal within experimental error.

A8- Figure 9a in fact shows a slight decrease of the HCl accommodation coefficient on beta-NAT similar to alpha-NAT (Figure 8a) where the decrease is a little larger over a similar T-range. However, as the referee suggests it may or may not be significant for beta-NAT.

TXT- . . .decreases as a function of temperature in the range 177-201 K, varying from 0.025 at 177 K to 0.016 at 201 K which may or may not be significant.

Q9- Page 19, lines 563-574 are the two distinct temperature regimes in Figure 2a due to surface disorder on ice?

A9- We certainly suggest this to be due to contamination-induced surface disorder that is discussed in the next few paragraphs and that has been highlighted in the studies of McNeill et al. However, at this point this remains a suggestion because we do not have structural proof of this hypothesis because in the present case the term "multidiagnostic" does not extend the investigation to structural studies.

Q10- Page 24, lines 704-709 why only TO experiments were possible for HNO3? This point is not so clear.

A10- The answer to this question has been given in Section 2.2, line 275-279.

Q11- Page 25, lines 753-758 can the authors comment why their results for HCl experiments were different from those by Haynes (2002)?

A11- We have the suspicion that the difference has to do with the fact that Hynes et al. (2002) performed their experiments at significantly higher temperatures which possible enables reversibility. This is mentioned on pg. 28, lines 834-837.

Q12- Figures 2-7 although the C2 authors mentioned the symbols in the text but it was so confusing to keep going back and forth between the text and the figure given the extra length of this manuscript and the different systems studied. I recommend that the authors explain the symbols in the caption for every figure.

A12- The captions have been written according to the guidelines of ACP. Owing to the complexity of the Figures we have added explanation of the symbols inside the Figures.

———————————————

[Figure]

Figure 1a (Caption in Report)

[Figure]

**Fig. 125** Transmission of silicon, thickness 2.5 mm. Dashed curve is for a sample coated to reduce reflection loss. [*From Texas Instruments (no date)*.]

Figure 1b (Caption in Report)

**Fig. 1.** Fig1a,1b and 2 for Answers to Referee1 and Referee2

---

## Author Comment (AC2) · 27 Jul 2016

General Comments:

Q- However, the organization and motivation needs to be made clearer, both in the introduction and in the atmospheric implications. Both sections read like a "data dump" with little explanation to identify the key discrepancies or limitations in the literature. Why are the authors conducting this study, 20+ years after some of the initial studies were conducted?

A- Referee 2 raises an important point: Why unfold the glory of heterogeneous chemistry once more (or once and for all?) after 20 years of (waning) interest? It may have

escaped the attention of Referee 2 that we report unique kinetic data secured by a consistency check (called thermochemical kinetics by the late S.W. Benson). There are NO available data in the literature on absolute rates of evaporation, not only for ice, but also for ices contaminated to various degrees by atmospheric trace gases. These data determine the evaporative lifetimes of various ice particles thought to be important in the UT/LS, and we have introduced a synoptic Table (Table 5 in the Discussion Section) in order to demonstrate the usefulness and the atmospheric importance of the kinetic data. Needless to say that we have made the point that in most cases the evaporative lifetimes enable heterogeneous processing to occur in a realistic time frame.

Why have 20 years gone by before coming forth with such seemingly important and useful data? The answer to this is multifactorial. It also has to do with the multidiagnostic capabilities of the present instrument that we have built up since 2003 in order to correct for the shortcomings of other experiments (Hanson and Ravishankara – single diagnostics flow tubes; Tolbert and coworkers – spectroscopy in chamber experiments, essentially w/o kinetics capabilities, Aerodyne group Chuck Kolb and Doug Worsnop performing single diagnostic equilibrium experiments for constructing phase diagrams, etc.). We have built an instrument with a decisive improvement in that it provides a unique spot of lowest temperature in a Stirred Flow Reactor w/o extraneous and uncontrolled cold spots that would perturb the reaction kinetics (through condensation of molecule of interest on an unidentified cold spot rather than on an optical support (FTIR, FTRAS, Quartz Crystal MicroBalance (QCMB), optical (HeNe) interferometry, etc.). We believe that the present measurements reveal hitherto unknown kinetic data at an unprecedented level of detail that are checked for mutual consistency by comparing the calculated equilibrium vapor pressure with known literature values.

The Introduction has been curtailed a bit in order to concentrate on the issues at hand. On the other hand, the paper has to be useful also for the non-specialist by providing at least the rudiments of a suitable atmospheric context. The impression of a "data dump"

is not wrong, except that it is sometimes unavoidable. We have made every effort to "lighten up" the text accordingly. Suffice it to say that we are proud and lucky to be able to present a manifold of hopefully useful data to the scientific community. More often than not papers seem to contain less than meets the eye, we think that we are in the contrary position of "more than meets the eye"!

Q- Instead, the intro leads with a nice (but unnecessary) review of general PSC chemistry, something that is now several decades old and the overview of which is not necessary.

A- See above paragraph in relation to presenting a self-contained account of atmospheric context.

Q- The atmospheric implications section goes on a tangent (incorrectly, at that) on water vapor measurement instruments that really aren't related to the current study results. Both of these aspects distract the reader from the high-quality, laboratory study and their results. While this paper will be eventually publishable, it requires some significant revisions in its current form.

We are heeding the advice of Referee 2 and have cut 90% of the material covering the Cryogenic Mirror Hygrometer. The only thing left is a brief description of the experiments of Gao et al. (2016) and the ramifications of the kinetic results of the present study.

Detailed (key) Points:

Q- Lines 56-129: a review of PSC chemistry has been common knowledge for decades; this section reads like a review article and is not necessary for the manuscript; indeed, it distracts from the critical questions that this study is trying to examine

A- In the interest of presenting a self-contained story we decided to keep this part in the Introduction.

Q- Lines 130-157: While this is a thorough review of the literature, it reads to some

extent like a "data dump". The experiments were done under different conditions to some extent. Are there real discrepancies between these results? Perhaps a table of past literature and your results would be more clear/helpful. At the very least, one should summarize the point of this section: e.g. there are discrepancies, there may or may not, too hard to say given the different experimental conditions, etc. and whatever it is, this is the motivation for our study! As written, the reader is left to search through a lot of data with no clear idea on whether there is true disagreement or not. And then explicitly tell what aspect of the study will your work address in this regard.

A- We have summarized the planned experiments in lines 152-169 by emphasizing at the end the thermochemical as well as the mass balance aspect which are the two novel aspects that make our measurements unique. On the other hand, we have refrained from evaluating the disparate kinetic results in the literature mentioned briefly on lines 116-149 that collect all relevant literature results to date. It is incumbent on reviews such as JPL and IUPAC rather than on original research papers to evaluate kinetic results of atmospheric importance.

TXT- In addition, all experiments have been performed under strict mass balance control by considering how many molecules of HNO3, HCl and H2O were present in the gas vs. the condensed phase (including the vessel walls) at any given time. These experiments have been described by Iannarelli and Rossi (2015). Most importantly, the consistency of the accommodation and evaporation kinetics has been checked using the method of thermochemical kinetics (Benson, 1976) by calculating the equilibrium vapor pressure and comparing it with values of published phase diagrams. In addition, the present work is the first to present absolute rates of evaporation of all involved constituents (H2O, HNO3, HCl) thus enabling predictions on evaporative lifetimes of ice particles under atmospheric conditions.

Q- Paragraph 158-163: Now suddenly the authors switch their literature review to HNO3 on pure ice. Only the last two sentences of this paragraph seem relevant to the work, at least for the introduction. And even then, there should be a transitional

statement such as "The complications/discrepancies of HNO3 and H2O update on NAT surfaces is also evident when examining HCl uptake on NAT." or similar.

A- As you guess the single component uptake kinetics of HCl and HNO3 on pure ice are also important when discussing uptake on binary chemical systems such as HNO3/H2O (this work) or HCl/H2O (Iannarelli and Rossi, 2013).

TXT- In the investigation of the properties of binary chemical systems the behavior of the simple single-component systems is an important stepping stone.

Q- Line 192+: The authors mention that the inlet system was modified but then failed to even provide a brief sentence or two on the actual modification. If it is important to mention at the start, please briefly summarize the modification.

A- Done

TXT- We therefore minimized the volume of the admission system and only retained the absolutely necessary total pressure gauge for measuring the absolute inlet flow rate (molecule s-1).

Q- Line 222-224: Not entirely an apples-to-apples comparison. The RAIR study was most sensitive to very thin films (< 10s-100 nm) and the very near surface properties. At thicker films and higher dose rates, they observed similar results as past studies and even the current study in the manuscript. The technique in the manuscript most likely was not sensitive to the presence of very thin films that were observed in the RAIR study.

A- We intended from the outset to avoid thick films owing to kinetic complications. Very often thin films occur as islands on the substrate or on the ice film such that the kinetics are ill-defined. Therefore, we chose to study the binary systems as thick films using the absorption cross sections that we have measured on thick films.

Q- Line 236-239: This is one of my main concerns experimentally about the study. The excess ice, even if it stabilizes NAT, will impact the vapor pressures of water inside the

chamber. Is one always on the ice-NAT phase line? If not, to what extent does each phase determine the partial pressure of water observed in the chamber? This has implications for the later results for the accuracy of H2O partial/vapor pressures and H2O kinetics. Why not just make a pure NAT film like to past thin film studies? NAT is a pretty stable film to make. The excess ice phase present needs to be discussed in more detail and a logical, reasoned argument why it doesn't complicate the interpretation of the results (or to what extent it does).

A-Figure 2 (introduced as supplemental Figure S5 into the SI section) presents a phase diagram of the binary system HNO3/H2O. According to Gibb's Phase Rule we have two components and three phases leading to a single degree of freedom. The dashed lines are isotherms, and as long you keep T constant you see that the equilibrium vapor pressure Pvap of H2O or HNO3 change within one and 3.5 orders of magnitude, respectively, depending on the composition (mass) of both solid phases, either H2O or HNO3 rich. The symbols (red for alpha-, black for beta-NAT) represent experiments characterized by a given value of PHNO3 and PH2O depending on the evaporation history of the ice sample. You also see the parameter space for the polar lower stratosphere and the number of points falling into the corresponding phase space of NAT. Riding the coexistence line is only interesting for the construction of the phase diagram in case it is not known. From Figure 2 you can read off both H2O and HNO3 vapor pressures and conclude that the present experiments are indeed relevant for the UT/LS as far as the vapor pressures are concerned (see your question below).

INSERT Figure 2

Figure 2: Phase diagram for the HNO3/H2O system in the range of atmospheric interest. The phase diagram is taken from Chapter 2 "The Probable Role of Stratospheric "Ice" Clouds: Heterogeneous Chemistry of the "Ozone Hole" by M.J. Molina, "The Chemistry of the Atmosphere: Its Impact on Global Change", J. Calvert (ed.), IUPAC Chemrawn 21 Series, Blackwell Scientific Publications.

Q- Lines 624-633: Report the entropies of evaporation as well – do they make sense with physical principles? If not, why? And elsewhere in the manuscript.

A- Taking exp(DELTAS0ev/2.303R)= 10(13.8) after conversion from Torr into an atmosphere we obtain DELTAS0ev = 264.6 J K-1 mol-1 or 63.25 cal K-1 mol-1. If we make the assumption that all H2O comes from NAT we have to divide by three owing to the fact that the decomposition of the trihydrate liberates three moles of H2O. We therefore have a value of 0.333x63.25 cal K-1 mol-1 or 21.1 cal K-1 mol-1 which exactly corresponds to Trouton's rule. However, this may just be fortuitous, also because we have a multicomponent system with several phases, each with its own thermodynamic parameters as we have to contend with the T-dependence of the combined system. The reason we are not discussing entropies of vaporization in this context is that the temperature range over which the measurements were taken is too small to obtain a reliable intercept, or in other words, the extrapolation of 1/T ==> 0.0 is too uncertain given the measurement range. This uncertainty owing to extrapolation is much larger than any potential effects of hydrogen bonding of H2O, HNO3 or HCl which is known to affect Trouton's rule (towards an increase of Trouton's constant).

Q- Line 660-661: Could a similar explanation be used to invoke discrepancies between your results and those in the literature (JPL recommendations)?

A- We are not sure about your question. Which discrepancies and which JPL recommendations?

Q- Line 681: Can the absolute value of 12 kJ/mole be explained physically in terms of hydrogen bonds? Why or why not?

A- From the point of view of the numerical value 12 kJ/mole is approximately$\frac{1}{3}$ to $\frac{1}{4}$ of a "normal" hydrogen bond. However, this single value is difficult to interpret in the absence of other supporting values. However, we feel that it is related to the fact that ïĄ ̧-NAT is not the most stable form of NAT. This primarily concerns the arrangement of H2O in the lattice which becomes tighter in ïĄ ́-NAT and therefore stabilizes the solid

hydrate.

Q- Line 686-7: Warshawsky et al. GRL 1999 also quantified this process of a sealing NAT layer slowing ice evaporation, and these were done a much lower $HNO_3$ partial pressures than in the Biermann et al. study. Related to this, what are the partial pressures of $HNO_3$ used in these experiments? Are they relevant to the atmosphere at all? They seem like they were much higher than what is expected in the atmosphere based upon the discussion and comparison to other laboratory studies. Please cite the $HNO_3$ partial pressures used in these experiments.

A- Please see above in conjunction with the binary phase diagram displayed in Figure 2 and/or Figure S5.We would like to affirm that the present conditions indeed are relevant to the UT/LS atmosphere as indicated in Figure 2 above by the symbols. Thank you for the Warshawsky citation that I routinely take from Maggie Tolbert's review article in Annual Rev. Phys. Chem.

TXT- Rev($H_2O$) on both alpha-NAT and beta-NAT is smaller compared to Rev($H_2O$) on pure ice. This is in agreement with the results of Tolbert and Middlebrook (1990), Middlebrook et al. (1996), Warshawsky et al. (1999) and Delval and Rossi (2005) who showed that ice coated with a number of molecular layers of NAT evaporates $H_2O$ at a slower rate than pure ice. On the other hand, our results are in contrast with the findings of Biermann et al. (1998) who report that no significant decrease of the $H_2O$ evaporation rate was observed in $HNO_3$-doped ice films. The discrepancy may possibly be caused by the high total pressure of 0.85 mbar in their reactor compared to all other competitive studies cited above that use high-vacuum chambers with total pressures lower by typically a factor of 500 or more. It is very likely that the experiments performed by Biermann et al. (1998) were not sensitive to changes in evaporation rates despite the fact that both the $HNO_3$ and $H_2O$ concentrations used as well as the thickness of the accumulated NAT layers in their no. 5 experiment were of the same magnitude as in the competing studies. A hint to that effect is the unexpected time dependence of the ice evaporation rate in Biermann et al. (1998) that shows an induction

time of 30 minutes as opposed to the expected linear decrease from the beginning of evaporation (see below). We are unable to attribute the source of the measured H2O vapor in the presence of two H2O-containing solid phases in our chemical system, namely pure H2O ice and NAT. We restate that the partial pressures at constant temperature are controlled by the (relative) composition of the system in agreement with the single degree of freedom resulting from Gibb's Phase Rule and the data displayed in the binary HNO3/H2O phase diagrams displayed in Figures 3, 5 and S5.

Q- Atmospheric implications: There is a data dump of numbers here once again, many of which were already described in detail in the discussion section. What are the key points, circling back to the motivation in the introduction and past literature experiments? e.g. Does the JPL kinetic data need to be revised (as suggested in the discussion in several places)? What are the implications of these much different values? How has this study broadened the range of past studies or explained potential discrepancies or unanswered questions in the past literature? What future research is needed? etc.

A- Many of the questions raised by Referee 2 for the "Conclusions and Atmospheric Implications" Section (5) are out of scope for a publication providing fundamental kinetics and thermodynamic data. We are unable to tackle all the suggested questions and do not see it as our task to provide evaluations of rate data on behalf of the JPL or IUPAC panels because this activity is built on consensus. We are happy to provide the best available answers surrounding the HNO3 hydrates to date. However, we have added Table 5 that is a vivid example and illustration of the usefulness of the obtained data in an atmospheric context, namely absolute rates of evaporation.

TXT- A look at Table 5 reveals evaporative lifetimes of various ice particles with respect to H2O evaporation. Equation (26) and (27) present the rudiments of a very simple layer-by-layer molecular model used to estimate evaporation lifetimes ($\theta$tot) at atmospheric conditions (Alcala et al., 2002; Chiesa and Rossi, 2013): $\theta$tot = (r/a)NML/Jevrh (26) Jevrh = Jevmax(1-rh/100) (27) with r, a, rh and NML being the radius of the ice

particle, shell thickness, relative humidity in % and the number of molecules cm-2 corresponding to one monolayer. Jevrh and Jevmax are the evaporation fluxes of H2O at rh and rh = 0, the latter corresponding to the maximum value of Jev.which we calculate following Equation (2) or (8). The salient feature of this simple evaporation model is the linear rate of change of the radius or diameter of the particle, a well- and widely known fact in aerosol physics in which the shrinking or growing size (diameter) of an aerosol particle is linear with time if the rate of evaporation is zero order, that is independent of a concentration term. Table 5 lists the evaporation life times which are not defined in terms of an e-folding time when dealing with first-order processes. In this example the lifetime is the time span between the cradle and death of the particle, this means from a given diameter 2r and "death" at 2r = 0. The chosen atmospheric conditions correspond to 190 K, rh = 80%, a = 2.5 Å for H2O and 3.35 Å for all other systems, r = 10 micro m and estimated values 6 x 1014, 3 x 1014 and 1 x 1015 molec cm-2 for NML of HNO3, HCl and H2O. It is immediately apparent that there is a large variation of $\theta$tot values for atmospherically relevant conditions which goes into the direction of increasing opportunities for heterogeneous interaction with atmospheric trace gases, even for pure ice (PSC type II). Table 5 is concerned with the most volatile component, namely H2O. If we now turn our attention to the least volatile component such as HNO3 in beta-NAT we obtain $\theta$tot = 5.1 d and 33.9 d for 0 and 85% HNO3 atmospheric saturation, the former being the maximum possible evaporation rate for 0% HNO3 saturation. The other boundary conditions are 190 K, polar upper tropospheric conditions at 11 km altitude (226.3 mb at 210 K), 1 ppb HNO3, 10 ppm H2O corresponding to 85% HNO3 saturation. This goes to show that laboratory experiments on gas-condensed phase exchange of lower volatility components in atmospheric hydrates are fraught with complications. It follows as a corollary that both HCl, but especially HNO3 contamination of H2O ice is bound to persist for all practical atmospheric conditions.

Q- Also, the discussion on NAT-coated ice impacting field measurements is speculative, unsupported, and shows several large gaps of awareness in UTLS water vapor measurements. First, the authors cite the problems of "reliable and reproducible measurements" of water vapor in the field UTLS measurements. However, as noted above, I have serious questions on how one can reliably interpret the accuracy of the water vapor measurements in their laboratory setup given that two phases exist at conditions well off the ice/NAT equilibrium line (and higher $HNO_3$ partial pressures than usually exist in the UTLS) – so it isn't clear to me how these laboratory results are that representative of the UTLS itself. Second, the CU/NOAA chilled mirror hygrometer has a long measurement history and is best described most recently by the Vömel et al. JGR, 2007 and/or Vömel et al., AMTD, 2016. More importantly, it compared extremely well in recent intercomparison campaigns to the reference standard (see Fahey et al. AMT 2014), an instrument/technique that probably is (in this reviewers' opinion) the most accurate/uncertainty-documented $H2O$ measurement in the community. Third, $HNO_3$ has not been shown in the NOAA tests to impact the frost layer (ice vapor pressure) at relevant $HNO_3$ concentrations (Thornberry et al., AMT 2011). Fourth, there are numerous diode laser-based hygrometers by many leading groups in the world; in fact, I would argue the NOAA TDL is one of the most recent and, though promising and a quality measurement, has some of the least amount of field data to characterize its strengths and weaknesses. More recent AquaVIT2 UT/LS water vapor intercomparisons showed some improved agreement in general from most of the UTLS hygrometers, whether diode laser-based at any wavelength (1.3, 1.4, 2.6 microns), laser-induced fluorescence, chilled mirrors, or other techniques. Therefore, I'm not sure the authors' results are applicable to explaining whether or not an instrument may work with the limited knowledge of the measurement instruments themselves and better agreement now being observed. This is especially true since the manuscript's lab results appear to be at $HNO_3$ concentrations/thicknesses well above what is possible in the UTLS. The Gao et al. 2016 JPC-A dealt with very small amounts of residual $HNO_3$ within ice and not related to thick NAT coatings here. For all of these reasons, I suggest removing these paragraphs on $H2O$ measurements and expanding on the kinetics and the implications thereof/discrepancies.

A- First we agree with Referee 2 that we are in no way specialists in the question of

H2O vapor measurements under UT/LS conditions. We therefore take out this section entirely and only mention the Gao et al. (2016) measurements at the very end as they directly relate to the present kinetic results inasmuch as the persistence of the lower volatility components in ice, namely HNO3, is concerned. We are a bit surprised at the explicit reaction of Referee 2 concerning the atmospheric relevance of the present study. We resolutely take exception to his statements to be "well off the ice/NAT equilibrium line (and higher HNO3 partial pressures than usually exist in the UTLS)". Figure 2 (S5 in the SI Section) clearly points out that (1) the UT/LS conditions are in the middle, not the limits of the NAT existence area within the relevant phase diagram, and that (2) the HNO3 partial pressure are not higher than usually exist in the UT/LS region. If anything, they are a bit lower because we have emphasized lower temperatures. In addition, we assert in contrast to Referee 2 that the NAT layers, typically 300 nm or less thick in the present study, are well representative of "what is possible in the UTLS"! In the end we consider it wise to continue to question measurement concepts for field applications using fundamental research instruments and methods. It is incumbent on us active in the laboratory to alert field scientists to possible shortcomings and artifacts of routinely applied methods and techniques used in the field.
* * *
[Figure]

[Figure]

Figure 2 (for Answers to Referee 2)

**Fig. 1.**

---

## Author Comment (AC3) · 27 Jul 2016

Dear Colleague,

We thank you very much for the detailed report on our paper dealing with kinetic and thermodynamic aspects of binary and ternary hydrates of HNO3 and HCl under UT/LS conditions. As advised by Referee 2 we have decided to remove the entire section on the background section of the Cryogenic Mirror Hygrometer (CMH) as we recognize that we do not have sufficient expertise in this area. However, we have written a new section at the end of the "Conclusions and Atmospheric Implications" Section (5) focusing on your newest experiments (Gao et al., 2016) and how it fits together with the kinetic results of the present study. We have been guided by the succinct description of

your experiment in your Interactive Comment that helped us to define and sharpen the salient points of your experiment. We have come to the conclusion that your observations are entirely consistent with the rates of evaporation of HNO3 and H2O measured in the present study when we extrapolate our rates to the temperature range of interest in your newest study, namely 207-213 K. We therefore do not have a point of contention with the conclusions of above-referenced paper anymore and thought it to be worthwhile to point this out at the end of our report even though we cannot make a guess as to the nature of the "second condensate" that must await further investigations.

Below you will find the significant insert into the text of the present paper:

At last it is useful to view the outcome of a recent laboratory experiment dealing with the binary HNO3/H2O system monitored using a cryogenic mirror hygrometer (CMH) (Gao et al., 2016) in light of the present kinetic results. In the basic experimental set-up the behavior of the sample CMH exposed to a combined low pressure H2O/HNO3 flow is compared to the response of a reference CMH that is located upstream of the HNO3 source and exposed to the H2O flow alone, and has been described in detail by Thornberry et al. (2011). Any increase in scattering of the incident monitoring laser beam owing to growth of the polycrystalline ice deposit will be counterbalanced by heating of the mirror to bring back the optical detector signal to a predetermined set point. The typical experimental sequence in Gao et al. (2016) starts by establishing pure ice frost layers on both CMH mirrors at a stable mixing ratio of < 10 ppm after which a continuous flow of HNO3 was added such that the flow past the sample CMH contained 80-100 ppb HNO3.

After typically one hour the gradual build-up of a NAT layer on the CMH was accompanied by a temperature increase of the sample CMH to settle around the saturation temperature Tsat of NAT at the chosen H2O and HNO3 flow rate. An increase of the H2O flow from 6 to 80 ppm led to ice growth on both mirrors accompanied by an increase of Tsat of NAT adjusting to the new H2O flow rate. Suddenly, the HNO3 flow was shut off which first led to a rapidly decreasing MS signal for HNO3 but ending up in

an above background signal corresponding to 0.5 to 1.0 ppb HNO3. The temperature of the sample CMH continued to decrease below Tsat of pure ice monitored by the reference CMH suggesting that Peq(H2O) of the condensate had become larger than that of pure ice. This solid state on the sample CMH was called "second condensate". The low level of HNO3 continued to react to repetitive increases (CMH heating) and decreases (CMH cooling) of the H2O flow in a reproducible manner all the while staying below the level corresponding to Tsat of pure ice on the reference CMH. These repetitive H2O on-off sequences provided additional evidence of the continued evaporation of HNO3 from the condensate. The response of HNO3 leaving the condensate undersaturated with respect to NAT is at first sight certainly unexpected based on the results displayed in Figures 2b and 4b. However, if one considers the relatively high mirror temperatures ranging between 207 and 213 K between which the "second condensate" was cycled by way of changing the H2O flows it suddenly becomes conceivable that Rev(HNO3) becomes equal to Rev(H2O) in that temperature range. Linear extrapolation of the absolute rates of evaporation hints at similar magnitude for temperatures exceeding 210 K beta- NAT (Figure 4). For alpha-NAT the temperature at which the evaporation rates of H2O and HNO3 become equal is even below 200 K owing to a steeper T-dependence of Rev(HNO3) in alpha-NAT (Figure 2 and Table 4). We conclude, that the observed dynamics of the experiment performed by Gao et al. (2016) is entirely consistent with the kinetic results of the present study. However, the results of the Gao et al. (2016) laboratory experiment would certainly be different at lower temperatures more representative of the UT/LS.

---

## Author Response (AR1)

**Answers to Question of Referee 1:**

Questions asked by referee is in straight font, *answers by the authors are given in ITALICS after the corresponding Question*. Modified text is given in small straight font in RED. In order to facilitate the location of text and/or Figures and table additions the reader will find a "Marked Copy" in "Track mode" where added text, Figures and Tables may be found suitably marked.

Questions/*Answers*/Modified or Added Text:

2- Page 7, line 185, silicon has a cutoff of 1500 cm$^{-1}$ in the FTIR so how can the range extend from 4000-700 cm-1?

[Figure]

Fig. 1a: Absorption spectrum of Si window (commercially available material from Nicodom sro)

[Figure]

**Fig. 125** Transmission of silicon, thickness 2.5 mm. Dashed curve is for a sample coated to reduce reflection loss. [*From Texas Instruments (no date).*]

Fig. 1b: Taken from the Handbook of Optics (Optical Society of America, McGraw-Hill book

Co. 1978)

*Figures 1a and 1b present transmission curves of Si windows that always have a very thin*

*coating (on the order of 50-100 nm) $SiO_2$ that protects the bulk of Si from oxidation. Although*

*transmission is reduced in the 1500 to 600 nm range it is sufficiently transmitting to enable*

*high-quality absorption spectra to be recorded. In our case the DIGILAB FTS 575 provides*

*high throughput thanks to its 3" collection optics. The centerburst signal reduces from 9V to*

*3V after passage across 2a pair of 5mm thick KCl and a 2.0 mm thick Si window with*

*external location of the HgCdTe detector cooled at 77 K.*

3- Page 8, lines 219-220, the authors discuss that the transition in phases was observed via
FTIR yet no FTIR or MS spectra were shown in the entire 52 pages of the manuscript. It
would be interesting to the readers to show sample spectra and also to mention in a table the
m/z and the wavenumbers where hydrates, $HNO_3$, HCl and water were observed.

*We agree with the referee regarding the presentation of raw FTIR/MS data of the discussed*

*ternary $HNO_3/HCl/H_2O$ chemical systems. To this effect we have added two new Figures (6*

*and 7) displaying combined FTIR/MS sample data as well as corresponding Table 3.*

*However, for the binary system $HNO_3/H_2O$ we have presented the corresponding combined*

*sample FTIR absorption/MS data already in the Iannarelli and Rossi (2015) publication (J.*

*Geophys. Res. 120, 11707-11727, 2015) such that renewed presentation in the present context*

*would appear not to be appropriate. We therefore point out this reference when discussing*

*the thermodynamic and kinetic data of the simple binary $HNO_3/H_2O$ system.*

We refrain at this point from showing raw data (FTIR absorption spectra and MS data as a function of time) because representative samples have been shown by Iannarelli and Rossi (2015) for α- and β-NAT. We will defer the presentation of raw data on the interaction of HCl on α- and β-NAT to Section 3.3 below.

*The following text is introduced into chapter 3.3 which introduces the ternary HCl/HNO₃/H₂O system.*

[revised manuscript text omitted]

4- Page 13, lines 369-372, the authors discussed the difference between Alpha-NAT and HCl; yet no HCl results were shown in Figure 2.

*The purpose of that statement regarding the difference between $R_{ev}(H_2O)$ in the HCl vs. the*
*$HNO_3$ hydrate was to alert the reader to a significant difference between the two hydrates. We*
*have inserted the two references that deal with the HCl hydrates (amorphous HCl hydrate*
*and HCl Hexahydrate).*

This result is very different compared to the previously studied case of HCl amorphous and crystalline hexahydrate using the same apparatus
(Iannarelli and Rossi, 2013), where the evaporation of $H_2O$ takes place at a rate characteristic of pure ice despite the presence of adsorbed
HCl on the ice and is in agreement with the findings of Delval and Rossi (2005).

5- Page 14, line 421, can the authors comment how the relative errors were calculated and
why same error in PV (30%) and TO (60%) experiments were observed on both the NAT and
NAD films?

*Although preferred from the point of view of avoiding sample saturation, we attribute twice*
*the uncertainty to the TO compared to the PV technique. TO involves taking a difference of*
*two large numbers in the denominator of Equations (7) and (8), which is the reason to*
*attribute a larger experimental uncertainty to this method.*

The largest uncertainty in our experiment is that of the flow rate introduced into the reactor, which is assigned a relative error of 25%. The
flow rate measurement affects the calibration of the MS and therefore the measurement of all the concentrations in the reactor (Eq. 4).
Therefore, we estimate a global relative error of 30% for PV experiments and double this uncertainty for TO experiments because Equations
(7) and (8) imply a difference of two large numbers in many cases, as discussed above. We therefore assign a global 60% relative error to
results obtained in TO experiments.

6- Page 15, lines 448-453, again the authors talk about comparisons to HCl experiments
however no HCl data are present in Figure 4b. Which figure the authors want the reader to
check to compare HCl case to figure 4a, please mention the figure since HCl experiments are
introduced in the next Section.

*As discussed above for alpha-NAT we are referring to a previous study on the BINARY*
*$HCl/H_2O$ phase (Iannarelli and Rossi, 2013) whereas chapter 3.3 below deals with the*
*TERNARY $HCl/HNO_3/H_2O$ system.*

As in the case of α-NAT, this result is very different compared to the case of HCl hydrates studied before using the same apparatus
(Iannarelli and Rossi, 2013) where the evaporation of $H_2O$ is not influenced by the presence of adsorbed HCl on the ice and takes place at a
rate characteristic of pure ice for all HCl concentrations used.

7- Page 17, lines 484-488, why are the authors making assumptions regarding the substrates
can't they get information on changes due to HCl from FTIR?

*In response to your discussion point 3 above we have introduced Figures 6 and 7 displaying*
*FTIR absorption spectra in the presence of HCl whose principal peak positions have been*
*collected in the new Table 3 (not reproduced here but included in the new manuscript*
*version). Regarding the ternary $HCl/HNO_3/H_2O$ system treated here we had to make some*
*verified assumptions in order to keep the experimental parameter space to an acceptable*
*level. All three simplifying assumptions have been verified in the current laboratory*
*experiments.*

In order to restrain the number of independent measurements on this ternary system to a practical level we had to make some assumptions
and/or simplifications in order to measure the unknown parameters of Eq. (2) for each gas used. Specifically, we made the following
reasonable assumptions, both for α-NAT and β-NAT substrates which have been experimentally verified in laboratory experiments:

8- Page 18, lines 534-535, the authors mentioned a decrease in ïA¸a¸-ïA¸c´-NAT as a function of increasing temperature but looking at figure 7a it looks like there was no change in the signal within experimental error.

*Figure 9a in fact shows a slight decrease of the HCl accommodation coefficient on β-NAT similar to α-NAT (Figure 8a) where the decrease is a little larger over a similar T-range. However, as the referee suggests it may or may not be significant for β-NAT.*

…decreases as a function of temperature in the range 177-201 K, varying from 0.025 at 177 K to 0.016 at 201 K which may or may not be significant.

9- Page 19, lines 563-574 are the two distinct temperature regimes in Figure 2a due to surface disorder on ice?

*We certainly suggest this to be due to contamination-induced surface disorder that is discussed in the next few paragraphs and that has been highlighted in the studies of McNeill et al. However, at this point this remains a suggestion because we do not have structural proof of this hypothesis because in the present case the term "multidiagnostic" does not extend the investigation to structural studies.*

10- Page 24, lines 704-709 why only TO experiments were possible for HNO3? This point is not so clear.

*The answer to this question has been given in Section 2.2, line 275-279.*

11- Page 25, lines 753-758 can the authors comment why their results for HCl experiments were different from those by Haynes (2002)?

*We have the suspicion that the difference has to do with the fact that Hynes et al. (2002) performed their experiments at significantly higher temperatures which possible enables reversibility. This is mentioned on pg. 28, lines 834-837.*

12- Figures 2-7 although the C2 authors mentioned the symbols in the text but it was so confusing to keep going back and forth between the text and the figure given the extra length of this manuscript and the different systems studied. I recommend that the authors explain the symbols in the caption for every figure.

*The captions have been written according to the guidelines of ACP. Owing to the complexity of the Figures we have added explanation of the symbols inside the Figures.*

**Answers to Question of Referee 2:**

Questions asked by referee is in straight font, *answers by the authors are given in ITALICS after the corresponding Question.* Modified text is given in small straight font in RED. In order to facilitate the location of text and/or Figures and table additions the reader will find a "Marked Copy" in "Track mode" where added text, Figures and Tables may be found suitably marked.

General Comments:

However, the organization and motivation needs to be made clearer, both in the introduction
and in the atmospheric implications. Both sections read like a "data dump" with little
explanation to identify the key discrepancies or limitations in the literature. Why are the
authors conducting this study, 20+ years after some of the initial studies were conducted?

*Referee 2 raises an important point: Why unfold the glory of heterogeneous chemistry once*
*more (or once and for all?) after 20 years of (waning) interest? It may have escaped the*
*attention of Referee 2 that we report unique kinetic data secured by a consistency check*
*(called thermochemical kinetics by the late S.W. Benson). There are NO available data in the*
*literature on absolute rates of evaporation, not only for ice, but also for ices contaminated to*
*various degrees by atmospheric trace gases. These data determine the evaporative lifetimes of*
*various ice particles thought to be important in the UT/LS, and we have introduced a synoptic*
*Table (Table 5 in the Discussion Section) in order to demonstrate the usefulness and the*
*atmospheric importance of the kinetic data. Needless to say that we have made the point that*
*in most cases the evaporative lifetimes enable heterogeneous processing to occur in a*
*realistic time frame.*

*Why have 20 years gone by before coming forth with such seemingly important and useful*
*data? The answer to this is multifactorial. It also has to do with the multidiagnostic*
*capabilities of the present instrument that we have built up since 2003 in order to correct for*
*the shortcomings of other experiments (Hanson and Ravishankara – single diagnostics flow*
*tubes; Tolbert and coworkers – spectroscopy in chamber experiments, essentially w/o kinetics*
*capabilities, Aerodyne group Chuck Kolb and Doug Worsnop performing single diagnostic*
*equilibrium experiments for constructing phase diagrams, etc.). We have built an instrument*
*with a decisive improvement in that it provides a unique spot of lowest temperature in a*
*Stirred Flow Reactor w/o extraneous and uncontrolled cold spots that would perturb the*
*reaction kinetics (through condensation of molecule of interest on an unidentified cold spot*
*rather than on an optical support (FTIR, FTRAS, Quartz Crystal MicroBalance (QCMB),*
*optical (HeNe) interferometry, etc.). We believe that the present measurements reveal hitherto*
*unknown kinetic data at an unprecedented level of detail that are checked for mutual*
*consistency by comparing the calculated equilibrium vapor pressure with known literature*
*values.*

*The Introduction has been curtailed a bit in order to concentrate on the issues at hand. On the*
*other hand, the paper has to be useful also for the non-specialist by providing at least the*
*rudiments of a suitable atmospheric context. The impression of a "data dump" is not wrong,*
*except that it is sometimes unavoidable. We have made every effort to "lighten up" the text*
*accordingly. Suffice it to say that we are proud and lucky to be able to present a manifold of*
*hopefully useful data to the scientific community. More often than not papers seem to contain*
*less than meets the eye, we think that we are in the contrary position of "more than meets the*
*eye"!*

Instead, the intro leads with a nice (but unnecessary) review of general PSC chemistry,
something that is now several decades old and the overview of which is not necessary.

*See above paragraph in relation to presenting a self-contained account of atmospheric*
*context.*

The atmospheric implications section goes on a tangent (incorrectly, at that) on water vapor
measurement instruments that really aren't related to the current study results. Both of these aspects distract the reader from the high-quality, laboratory study and their results. While this
paper will be eventually publishable, it requires some significant revisions in its current form.

*We are heeding the advice of Referee 2 and have cut 90% of the material covering the*
*Cryogenic Mirror Hygrometer. The only thing left is a brief description of the experiments of*
*Gao et al. (2016) and the ramifications of the kinetic results of the present study.*

Detailed (key) Points:

Lines 56-129: a review of PSC chemistry has been common knowledge for decades; this
section reads like a review article and is not necessary for the manuscript; indeed, it distracts
from the critical questions that this study is trying to examine.

*In the interest of presenting a self-contained story we decided to keep this part in the*
*Introduction.*

Lines 130-157: While this is a thorough review of the literature, it reads to some extent like a
"data dump". The experiments were done under different conditions to some extent. Are there
real discrepancies between these results? Perhaps a table of past literature and your results
would be more clear/helpful. At the very least, one should summarize the point of this
section: e.g. there are discrepancies, there may or may not, too hard to say given the different
experimental conditions, etc. and whatever it is, this is the motivation for our study! As
written, the reader is left to search through a lot of data with no clear idea on whether there is
true disagreement or not. And then explicitly tell what aspect of the study will your work
address in this regard.

*We have summarized the planned experiments in lines 152-169 by emphasizing at the end the*
*thermochemical as well as the mass balance aspect which are the two novel aspects that make*
*our measurements unique. On the other hand, we have refrained from evaluating the*
*disparate kinetic results in the literature mentioned briefly on lines 116-149 that collect all*
*relevant literature results to date. It is incumbent on reviews such as JPL and IUPAC rather*
*than on original research papers to evaluate kinetic results of atmospheric importance.*

In addition, all experiments have been performed under strict mass balance control by considering how many molecules of $HNO_3$, $HCl$ and
$H_2O$ were present in the gas vs. the condensed phase (including the vessel walls) at any given time. These experiments have been described
by Iannarelli and Rossi (2015). Most importantly, the consistency of the accommodation and evaporation kinetics has been checked using the
method of thermochemical kinetics (Benson, 1976) by calculating the equilibrium vapor pressure and comparing it with values of published
phase diagrams. In addition, the present work is the first to present absolute rates of evaporation of all involved constituents ($H_2O$, $HNO_3$,
$HCl$) thus enabling predictions on evaporative lifetimes of ice particles under atmospheric conditions.

Paragraph 158-163: Now suddenly the authors switch their literature review to HNO3 on pure
ice. Only the last two sentences of this paragraph seem relevant to the work, at least for the
introduction. And even then, there should be a transitional statement such as "The
complications/discrepancies of HNO3 and H2O update on NAT surfaces is also evident when
examining HCl uptake on NAT." or similar.

*As you guess the single component uptake kinetics of HCl and $HNO_3$ on pure ice are also*
*important when discussing uptake on binary chemical systems such as $HNO_3/H_2O$ (this work)*
*or $HCl/H_2O$ (Iannarelli and Rossi, 2013).*

In the investigation of the properties of binary chemical systems the behavior of the simple single-component systems is an important
stepping stone.

Line 192+: The authors mention that the inlet system was modified but then failed to even
provide a brief sentence or two on the actual modification. If it is important to mention at the
start, please briefly summarize the modification.

*Done*

We therefore minimized the volume of the admission system and only retained the absolutely necessary total pressure gauge for measuring
the absolute inlet flow rate (molecule $s^{-1}$).

Line 222-224: Not entirely an apples-to-apples comparison. The RAIR study was most
sensitive to very thin films (< 10s-100 nm) and the very near surface properties. At thicker
films and higher dose rates, they observed similar results as past studies and even the current
study in the manuscript. The technique in the manuscript most likely was not sensitive to the
presence of very thin films that were observed in the RAIR study.

*We intended from the outset to avoid thick films owing to kinetic complications. Very often*
*thin films occur as islands on the substrate or on the ice film such that the kinetics are ill-*
*defined. Therefore, we chose to study the binary systems as thick films using the absorption*
*cross sections that we have measured on thick films.*

Line 236-239: This is one of my main concerns experimentally about the study. The excess
ice, even if it stabilizes NAT, will impact the vapor pressures of water inside the chamber. Is
one always on the ice-NAT phase line? If not, to what extent does each phase determine the
partial pressure of water observed in the chamber? This has implications for the later results
for the accuracy of H2O partial/vapor pressures and H2O kinetics. Why not just make a pure
NAT film like to past thin film studies? NAT is a pretty stable film to make. The excess ice
phase present needs to be discussed in more detail and a logical, reasoned argument why it
doesn't complicate the interpretation of the results (or to what extent it does).

*Figure 2 (introduced as supplemental Figure S5 into the SI section) presents a phase diagram*
*of the binary system $HNO_3$/$H_2O$. According to Gibb's Phase Rule we have two components*
*and three phases leading to a single degree of freedom. The dashed lines are isotherms, and*
*as long you keep T constant you see that the equilibrium vapor pressure $P_{vap}$ of $H_2O$ or $HNO_3$*
*change within one and 3.5 orders of magnitude, respectively, depending on the composition*
*(mass) of both solid phases, either $H_2O$ or $HNO_3$ rich. The symbols (red for $\alpha$-, black for $\beta$-*
*NAT) represent experiments characterized by a given value of $P_{HNO3}$ and $P_{H2O}$ depending on*
*the evaporation history of the ice sample. You also see the parameter space for the polar*
*lower stratosphere and the number of points falling into the corresponding phase space of*
*NAT. Riding the coexistence line is only interesting for the construction of the phase diagram*
*in case it is not known. From Figure 2 you can read off both $H_2O$ and $HNO_3$ vapor pressures*
*and conclude that the present experiments are indeed relevant for the UT/LS as far as the*
*vapor pressures are concerned (see your question below).*

[Figure]

*Figure 2: Phase diagram for the HNO₃/H₂O system in the range of atmospheric interest. The*
*phase diagram is taken from Chapter 2 "The Probable Role of Stratospheric "Ice" Clouds:*
*Heterogeneous Chemistry of the "Ozone Hole" by M.J. Molina, "The Chemistry of the*
*Atmosphere: Its Impact on Global Change", J. Calvert (ed.), IUPAC Chemrawn 21 Series,*
*Blackwell Scientific Publications.*

Lines 624-633: Report the entropies of evaporation as well – do they make sense with
physical principles? If not, why? And elsewhere in the manuscript.

*Taking $exp(\Delta S^0_{ev}/2.303R)= 10^{13.8}$ after conversion from Torr into an atmosphere we obtain*
*$\Delta S^0_{ev} = 264.6\ JK^{-1}mol^{-1}$ or $63.25\ cal\ K^{-1}\ mol^{-1}$. If we make the assumption that all $H_2O$ comes*
*from NAT we have to divide by three owing to the fact that the decomposition of the trihydrate*
*liberates three moles of $H_2O$. We therefore have a value of $0.333x63.25\ cal\ K^{-1}\ mol^{-1}$ or 21.1*
*cal $K^{-1}\ mol^{-1}$ which exactly corresponds to Trouton's rule. However, this may just be*
*fortuitous, also because we have a multicomponent system with several phases, each with its*
*own thermodynamic parameters as we have to contend with the T-dependence of the*
*combined system. The reason we are not discussing entropies of vaporization in this context is*
*that the temperature range over which the measurements were taken is too small to obtain a*
*reliable intercept, or in other words, the extrapolation of $1/T \rightarrow 0.0$ is too uncertain given the*
*measurement range. This uncertainty owing to extrapolation is much larger than any*
*potential effects of hydrogen bonding of $H_2O$, $HNO_3$ or HCl which is known to affect*
*Trouton's rule (towards an increase of Trouton's constant).*

Line 660-661: Could a similar explanation be used to invoke discrepancies between your
results and those in the literature (JPL recommendations)?

*We are not sure about your question. Which discrepancies and which JPL recommendations?*

Line 681: Can the absolute value of 12 kJ/mole be explained physically in terms of hydrogen
bonds? Why or why not?

*From the point of view of the numerical value 12 kJ/mole is approximately1/3 to ¼ of a*
*"normal" hydrogen bond. However, this single value is difficult to interpret in the absence of*
*other supporting values. However, we feel that it is related to the fact that α-NAT is not the*

*most stable form of NAT. This primarily concerns the arrangement of $H_2O$ in the lattice which*
*becomes tighter in β-NAT and therefore stabilizes the solid hydrate.*

Line 686-7: Warshawsky et al. GRL 1999 also quantified this process of a sealing NAT layer
slowing ice evaporation, and these were done a much lower HNO3 partial pressures than in
the Biermann et al. study. Related to this, what are the partial pressures of HNO3 used in
these experiments? Are they relevant to the atmosphere at all? They seem like they were
much higher than what is expected in the atmosphere based upon the discussion and
comparison to other laboratory studies. Please cite the HNO3 partial pressures used in these
experiments.

*Please see above in conjunction with the binary phase diagram displayed in Figure 2 and/or*
*Figure S5.We would like to affirm that the present conditions indeed are relevant to the*
*UT/LS atmosphere as indicated in Figure 2 above by the symbols. Thank you for the*
*Warshawsky citation that I routinely take from Maggie Tolbert's review article in Annual*
*Rev. Phys. Chem.*

$R_{ev}(H_2O)$ on both α-NAT and β-NAT is smaller compared to $R_{ev}(H_2O)$ on pure ice. This is in agreement with the results of Tolbert and
Middlebrook (1990), Middlebrook et al. (1996), Warshawsky et al. (1999) and Delval and Rossi (2005) who showed that ice coated with a
number of molecular layers of NAT evaporates $H_2O$ at a slower rate than pure ice. On the other hand, our results are in contrast with the
findings of Biermann et al. (1998) who report that no significant decrease of the $H_2O$ evaporation rate was observed in $HNO_3$-doped ice
films. The discrepancy may possibly be caused by the high total pressure of 0.85 mbar in their reactor compared to all other competitive
studies cited above that use high-vacuum chambers with total pressures lower by typically a factor of 500 or more. It is very likely that the
experiments performed by Biermann et al. (1998) were not sensitive to changes in evaporation rates despite the fact that both the $HNO_3$ and
$H_2O$ concentrations used as well as the thickness of the accumulated NAT layers in their no. 5 experiment were of the same magnitude as in
the competing studies. A hint to that effect is the unexpected time dependence of the ice evaporation rate in Biermann et al. (1998) that
shows an induction time of 30 minutes as opposed to the expected linear decrease from the beginning of evaporation (see below). We are
unable to attribute the source of the measured $H_2O$ vapor in the presence of two $H_2O$-containing solid phases in our chemical system, namely
pure $H_2O$ ice and NAT. We restate that the partial pressures at constant temperature are controlled by the (relative) composition of the
system in agreement with the single degree of freedom resulting from Gibb's Phase Rule and the data displayed in the binary $HNO_3/H_2O$
phase diagrams displayed in Figures 3, 5 and S5.

Atmospheric implications: There is a data dump of numbers here once again, many of which
were already described in detail in the discussion section. What are the key points, circling
back to the motivation in the introduction and past literature experiments? e.g. Does the JPL
kinetic data need to be revised (as suggested in the discussion in several places)? What are the
implications of these much different values? How has this study broadened the range of past
studies or explained potential discrepancies or unanswered questions in the past literature?
What future research is needed? etc.

*Many of the questions raised by Referee 2 for the "Conclusions and Atmospheric*
*Implications" Section (5) are out of scope for a publication providing fundamental kinetics*
*and thermodynamic data. We are unable to tackle all the suggested questions and do not see*
*it as our task to provide evaluations of rate data on behalf of the JPL or IUPAC panels*
*because this activity is built on consensus. We are happy to provide the best available*
*answers surrounding the $HNO_3$ hydrates to date. However, we have added Table 5 that is a*
*vivid example and illustration of the usefulness of the obtained data in an atmospheric*
*context, namely absolute rates of evaporation.*

A look at Table 5 reveals evaporative lifetimes of various ice particles with respect to $H_2O$ evaporation. Equation (26) and (27) present the
rudiments of a very simple layer-by-layer molecular model used to estimate evaporation lifetimes ($θ_{tot}$) at atmospheric conditions (Alcala et
al., 2002; Chiesa and Rossi, 2013):

$$θ_{tot} = (r/a)N_{ML}/J_{ev}^{rh} \tag{26}$$

$$J_{ev}^{rh} = J_{ev}^{max}(1-rh/100) \tag{27}$$

with r, a, rh and $N_{ML}$ being the radius of the ice particle, shell thickness, relative humidity in % and the number of molecules $cm^{-2}$
corresponding to one monolayer. $J_{ev}^{rh}$ and $J_{ev}^{max}$ are the evaporation fluxes of $H_2O$ at rh and rh = 0, the latter corresponding to the maximum
value of $J_{ev}$.which we calculate following Equation (2) or (8). The salient feature of this simple evaporation model is the linear rate of change
of the radius or diameter of the particle, a well- and widely known fact in aerosol physics in which the shrinking or growing size (diameter)

of an aerosol particle is linear with time if the rate of evaporation is zero order, that is independent of a concentration term. Table 5 lists the
evaporation life times which are not defined in terms of an e-folding time when dealing with first-order processes. In this example the
lifetime is the time span between the cradle and death of the particle, this means from a given diameter 2r and "death" at 2r = 0. The chosen
atmospheric conditions correspond to 190 K, rh = 80%, a = 2.5 Å for $H_2O$ and 3.35 Å for all other systems, r = 10 µm and estimated values 6
x $10^{14}$, 3 x $10^{14}$ and 1 x $10^{15}$ molec $cm^{-2}$ for $N_{ML}$ of $HNO_3$, HCl and $H_2O$. It is immediately apparent that there is a large variation of $\theta_{tot}$
values for atmospherically relevant conditions which goes into the direction of increasing opportunities for heterogeneous interaction with
atmospheric trace gases, even for pure ice (PSC type II). Table 5 is concerned with the most volatile component, namely $H_2O$. If we now turn
our attention to the least volatile component such as $HNO_3$ in β-NAT we obtain $\theta_{tot}$ = 5.1 d and 33.9 d for 0 and 85% $HNO_3$ atmospheric
saturation, the former being the maximum possible evaporation rate for 0% $HNO_3$ saturation. The other boundary conditions are 190 K, polar
upper tropospheric conditions at 11 km altitude (226.3 mb at 210 K), 1 ppb $HNO_3$, 10 ppm $H_2O$ corresponding to 85% $HNO_3$ saturation. This
goes to show that laboratory experiments on gas-condensed phase exchange of lower volatility components in atmospheric hydrates are
fraught with complications. It follows as a corollary that both HCl, but especially $HNO_3$ contamination of $H_2O$ ice is bound to persist for all
practical atmospheric conditions.

Also, the discussion on NAT-coated ice impacting field measurements is speculative, unsupported, and shows several large gaps of awareness in UTLS water vapor measurements.

First, the authors cite the problems of "reliable and reproducible measurements" of water
vapor in the field UTLS measurements. However, as noted above, I have serious questions on
how one can reliably interpret the accuracy of the water vapor measurements in their
laboratory setup given that two phases exist at conditions well off the ice/NAT equilibrium
line (and higher HNO3 partial pressures than usually exist in the UTLS) – so it isn't clear to
me how these laboratory results are that representative of the UTLS itself. Second, the
CU/NOAA chilled mirror hygrometer has a long measurement history and is best described
most recently by the Vömel et al. JGR, 2007 and/or Vömel et al., AMTD, 2016. More
importantly, it compared extremely well in recent intercomparison campaigns to the reference
standard (see Fahey et al. AMT 2014), an instrument/technique that probably is (in this
reviewers' opinion) the most accurate/uncertainty-documented H2O measurement in the
community. Third, HNO3 has not been shown in the NOAA tests to impact the frost layer (ice
vapor pressure) at relevant HNO3 concentrations (Thornberry et al., AMT 2011). Fourth,
there are numerous diode laser-based hygrometers by many leading groups in the world; in
fact, I would argue the NOAA TDL is one of the most recent and, though promising and a
quality measurement, has some of the least amount of field data to characterize its strengths
and weaknesses. More recent AquaVIT2 UT/LS water vapor intercomparisons showed some
improved agreement in general from most of the UTLS hygrometers, whether diode laser-
based at any wavelength (1.3, 1.4, 2.6 microns), laser-induced fluorescence, chilled mirrors,
or other techniques. Therefore, I'm not sure the authors' results are applicable to explaining
whether or not an instrument may work with the limited knowledge of the measurement
instruments themselves and better agreement now being observed. This is especially true
since the manuscript's lab results appear to be at HNO3 concentrations/thicknesses well
above what is possible in the UTLS. The Gao et al. 2016 JPC-A dealt with very small
amounts of residual HNO3 within ice and not related to thick NAT coatings here. For all of
these reasons, I suggest removing these paragraphs on H2O measurements and expanding on
the kinetics and the implications thereof/discrepancies.

*First we agree with Referee 2 that we are in no way specialists in the question of $H_2O$ vapor*
*measurements under UT/LS conditions. We therefore take out this section entirely and only*
*mention the Gao et al. (2016) measurements at the very end as they directly relate to the*
*present kinetic results inasmuch as the persistence of the lower volatility components in ice,*
*namely $HNO_3$, is concerned. We are a bit surprised at the explicit reaction of Referee 2*
*concerning the atmospheric relevance of the present study. We resolutely take exception to his*
*statements to be "well off the ice/NAT equilibrium line (and higher $HNO_3$ partial pressures*
*than usually exist in the UTLS)". Figure 2 (S5 in the SI Section) clearly points out that (1) the*
*UT/LS conditions are in the middle, not the limits of the NAT existence area within the*

*relevant phase diagram, and that (2) the $HNO_3$ partial pressure are not higher than usually*
*exist in the UT/LS region. If anything, they are a bit lower because we have emphasized lower*
*temperatures. In addition, we assert in contrast to Referee 2 that the NAT layers, typically*
*300 nm or less thick in the present study, are well representative of "what is possible in the*
*UTLS"! In the end we consider it wise to continue to question measurement concepts for field*
*applications using fundamental research instruments and methods. It is incumbent on us*
*active in the laboratory to alert field scientists to possible shortcomings and artifacts of*
*routinely applied methods and techniques used in the field.*

[revised manuscript text omitted]